# Kastor and Polluks polypeptides encoded by a single gene locus cooperatively regulate VDAC and spermatogenesis

Shintaro Mise[1,10], Akinobu Matsumoto [1,10✉], Keisuke Shimada [2], Toshiaki Hosaka [3], Masatomo Takahashi [4], Kazuya Ichihara [1], Hideyuki Shimizu[1,5], Chisa Shiraishi[1], Daisuke Saito[6], Mikita Suyama [6], Tomoharu Yasuda[7], Toru Ide[8], Yoshihiro Izumi [4], Takeshi Bamba [4], Tomomi Kimura-Someya [3], Mikako Shirouzu [3], Haruhiko Miyata [2], Masahito Ikawa [2,9] & Keiichi I. Nakayama [1✉]

Although several long noncoding RNAs (lncRNAs) have recently been shown to encode small polypeptides, those in testis remain largely uncharacterized. Here we identify two sperm-specific polypeptides, Kastor and Polluks, encoded by a single mouse locus (*Gm9999*) previously annotated as encoding a lncRNA. Both Kastor and Polluks are inserted in the outer mitochondrial membrane and directly interact with voltage-dependent anion channel (VDAC), despite their different amino acid sequences. Male VDAC3-deficient mice are infertile as a result of reduced sperm motility due to an abnormal mitochondrial sheath in spermatozoa, and deficiency of both Kastor and Polluks also severely impaired male fertility in association with formation of a similarly abnormal mitochondrial sheath. Spermatozoa lacking either Kastor or Polluks partially recapitulate the phenotype of those lacking both. Cooperative function of Kastor and Polluks in regulation of VDAC3 may thus be essential for mitochondrial sheath formation in spermatozoa and for male fertility.

[1] Division of Cell Biology, Medical Institute of Bioregulation, Kyushu University, Fukuoka, Fukuoka 819-0395, Japan. [2] Department of Experimental Genome Research, Research Institute for Microbial Diseases, Osaka University, Suita, Osaka 565-0871, Japan. [3] Laboratory for Protein Functional and Structural Biology, RIKEN Center for Biosystems Dynamics Research, Yokohama, Kanagawa 230-0045, Japan. [4] Division of Metabolomics, Medical Institute of Bioregulation, Kyushu University, Fukuoka, Fukuoka 819-0395, Japan. [5] Department of AI Systems Medicine, M&D Data Science Center, Tokyo Medical and Dental University, Bunkyo-ku, Tokyo 113-8510, Japan. [6] Division of Bioinformatics, Medical Institute of Bioregulation, Kyushu University, Fukuoka, Fukuoka 819-0395, Japan. [7] Department of Immunology, Graduate School of Biomedical and Health Sciences, Hiroshima University, Hiroshima, Hiroshima 734-8551, Japan. [8] Graduate School of Interdisciplinary Science and Engineering in Health Systems, Okayama University, Okayama, Okayama 700-8530, Japan. [9] The Institute of Medical Science, The University of Tokyo, Bunkyo-ku, Tokyo 108-8639, Japan. [10]These authors contributed equally: Shintaro Mise, Akinobu Matsumoto. ✉email: akinobu@bioreg.kyushu-u.ac.jp; nakayak1@bioreg.kyushu-u.ac.jp

Many long noncoding RNAs (lncRNAs) are expressed in a tissue-specific manner in mammals, with the testis appearing to express the largest number of tissue-specific lncRNAs[1,2]. Testis-specific lncRNAs also manifest a stage-specific expression pattern during spermatogenesis, suggesting that they may have key roles in this process[3]. However, many lncRNAs have similar characteristics to mRNAs, such as being transcribed by RNA polymerase II, spliced, capped, and polyadenylated, and we and others have recently shown that some lncRNAs actually encode functional polypeptides[4–8]. Given the large number of testis-specific lncRNAs, it might be expected that some of these are currently misannotated and are actually polypeptide-coding RNAs.

Spermatogenesis consists of the steps of mitosis, meiosis, and spermiogenesis, resulting in the formation of spermatozoa from diploid spermatogonia[9]. Spermiogenesis is a phase of transformation in which round spermatids differentiate into spermatozoa through acrosome formation, nuclear condensation, flagellum development, and removal of unnecessary cytoplasmic components. During this step, the morphologically distinctive mitochondrial sheath is also formed. The sphere-shaped mitochondria in the cytoplasm thus accumulate and line up around the flagellum (alignment step), they extend laterally to form a crescent shape (interlocking step), and they then elongate further to form a double-helical structure (compaction step). As a result of these dynamic changes in their distribution and morphology, mitochondria become restricted to the midpiece of the spermatozoon and tightly wrapped around the flagellum, with this structure being known as the mitochondrial sheath[10,11]. Although loss of certain factors has been shown to result in abnormalities of mitochondrial sheath formation and in infertility[12–16], the detailed molecular mechanisms underlying such defects have remained largely unknown.

Voltage-dependent anion channel (VDAC) is a mitochondrial porin that localizes to the outer mitochondrial membrane (OMM) and is conserved in all eukaryotes[17]. Under conditions of high conductance, VDAC mediates the permeation of anions such as ATP and ADP, whereas under low-conductance conditions it preferentially mediates that of cations such as $K^+$, $Na^+$, and $Ca^{2+}$ (ref. [18]). The activity and function of VDAC are regulated by its binding to various proteins including adenine nucleotide translocator (ANT), Bcl-$x_L$, creatine kinase, hexokinase, tubulin, and α-synuclein, with such binding being required for voltage-sensitive reversible closure of VDAC as well as for mitochondrion-dependent apoptotic signaling and formation of the mitochondrial permeability transition pore (mPTP)[19–24]. Two and three members of the VDAC family have been identified in yeast (YVDAC1 and YVDAC2) and vertebrates (VDAC1, VDAC2, and VDAC3)[25], respectively, and the position of VDAC3 on a different branch of the phylogenetic tree suggests its function might differ from those of VDAC1 and VDAC2[26,27]. *Saccharomyces cerevisiae* deficient in YVDAC1 (*por1* mutant) manifests a temperature-sensitive growth defect in the presence of a nonfermentable carbon source (glycerol), and this defect is corrected by the introduction of human or mouse VDAC1 or VDAC2, and to a much lesser extent by that of VDAC3[27,28]. Moreover, VDAC3 shows weak voltage-dependent gating activity in artificial lipid bilayer systems[29], with such activity being triggered by suppression of disulfide-bond formation[28].

VDAC1, VDAC2, and VDAC3 are ubiquitously expressed in mouse tissues. A proportion of VDAC1-deficient mice die during embryogenesis, but surviving animals appear to be indistinguishable from wild-type (WT) mice and are fertile[30]. Mice deficient in VDAC2 are viable at birth, but they show impaired weight gain with age and their health status declines to a level that requires euthanasia by 6 weeks of age. In addition, male VDAC2-deficient mice have small testes that lack mature sperm but contain an increased number of spermatogonia[31]. VDAC3-null mice are viable and overall healthy, but males are infertile and possess spermatozoa with structural abnormalities of the mitochondrial sheath and axoneme[32]. Given the ubiquitous expression of VDAC3, it remains unclear why the phenotype of VDAC3 deficiency appears largely restricted to spermatozoa, but this specificity is suggestive of a spermatozoon-specific function and regulation of VDAC3.

In the present study, we screened testis-specific lncRNAs and identified two polypeptides, designated Kastor and Polluks, both of which are translated from the lncRNA Gm9999 in mouse. Kastor and Polluks were localized to the OMM and found to interact with VDAC isoforms, and male mice deficient in both Kastor and Polluks manifested markedly impaired fertility, with their spermatozoa showing abnormal formation of the mitochondrial sheath similar to that apparent in VDAC3-null spermatozoa. Kastor is expressed only during the middle steps of spermatid development, whereas Polluks is expressed from the late steps of spermatid development to the mature spermatozoa stage. Loss of either Kastor or Polluks resulted in abnormalities that were consistent with the timing of their expression. Our findings indicate that Kastor and Polluks play an essential role in VDAC3-dependent mitochondrial sheath formation, which is required for male fertility.

## Results

**Kastor and Polluks are expressed in testicular germ cells.** Given that the testis harbors the largest number of tissue-specific lncRNAs, we applied PhyloCSF, an algorithm for predicting evolutionarily conserved open reading frames (ORFs)[33], to screen 996 testis-specific lncRNAs of mouse in the NCBI database (Supplementary Data 1). This analysis revealed two potential ORFs within Gm9999, which showed a high RNA expression level and a high PhyloCSF score (Fig. 1a). Two mRNAs with different transcription start sites and different ORFs were thus predicted to be transcribed from the *Gm9999* locus (Fig. 1b). We termed the putative encoded polypeptides Kastor and Polluks, which comprise 53 and 40 amino acids, respectively (Fig. 1b). Both Kastor and Polluks were found to be conserved among mammals, but not in other vertebrates (Fig. 1c). Reverse transcription (RT) and polymerase chain reaction (PCR) analysis confirmed that transcripts encoding Kastor or Polluks were present only in testis among the tissues of adult mouse tested (Fig. 1d), whereas in situ hybridization analysis detected these transcripts at the step of spermiogenesis, with *Kastor* and *Polluks* transcripts showing slightly different expression patterns (Fig. 1e).

To evaluate the expression of endogenous Kastor and Polluks at the protein level, we generated two lines of knock-in mice that harbor a FLAG epitope tag sequence at the COOH-terminus of the respective polypeptide (*Kastor*FLAG/+ or *Polluks*FLAG/+ mice) with the use of the CRISPR–Cas9 system (Supplementary Fig. 1). Immunoblot analysis of lysates prepared from the testis or mature spermatozoa collected from the cauda epididymis confirmed the endogenous expression of both Kastor and Polluks (Fig. 1f, g). Polluks was detected in both testis and mature spermatozoa, whereas Kastor expression was observed only in testis. Moreover, we established monoclonal antibodies to mouse Kastor or Polluks, with which we further confirmed the endogenous expression of these proteins in testis or mature spermatozoa (Fig. 1h). Whereas the endogenous expression of Polluks was observed in the testis of *Polluks*FLAG/+ mice (Fig. 1f), it was not detected in the testis with the monoclonal antibodies (Fig. 1h), likely as a result of low sensitivity of the antibodies to Polluks as well as of the lower expression level of Polluks in the testis than in mature spermatozoa. Together, these results indicated that Gm9999, which was previously annotated as "noncoding," actually encodes the polypeptides Kastor and Polluks.

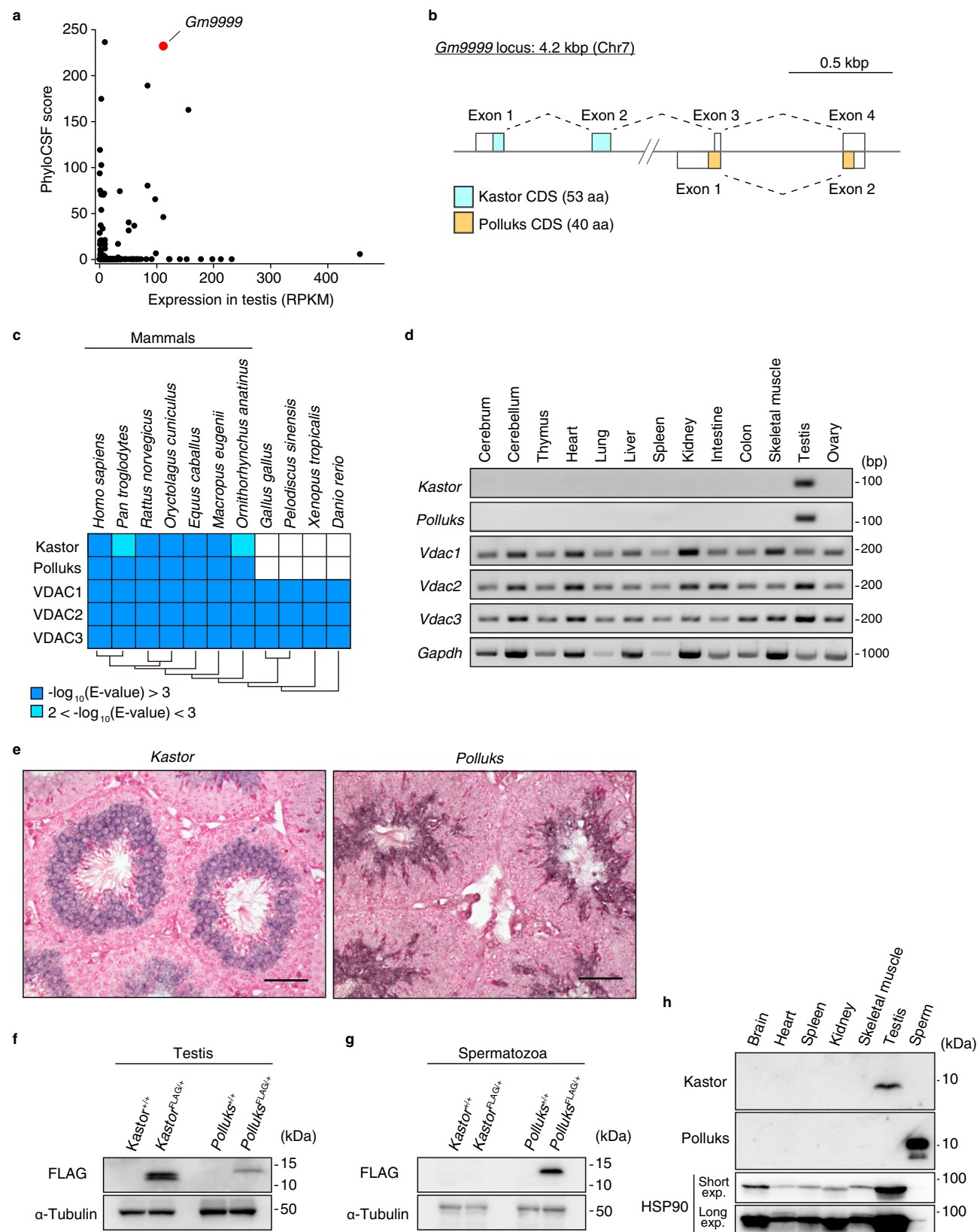

**Distinct expression of Kastor/Polluks during spermatogenesis.** We next examined the localization of Kastor and Polluks polypeptides in the testis by immunofluorescence analysis in *Kastor*<sup>FLAG/+</sup> and *Polluks*<sup>FLAG/+</sup> mice. Consistent with the results of in situ hybridization analysis (Fig. 1e), Kastor and Polluks were not detected in spermatogonia (E-cadherin positive) or spermatocytes (synaptonemal complex protein 3 [SYCP3] positive) or in Leydig cells (3β-hydroxysteroid dehydrogenase [3β-HSD] positive) or Sertoli cells

**Fig. 1 Gm9999 encodes Kastor and Polluks. a** PhyloCSF scores and expression levels of 996 testis-specific lncRNAs of mouse in the NCBI database. RPKM, reads per kilobase of transcript per million mapped reads. **b** Schematic representation of the *Gm9999* locus encoding Kastor and Polluks. Two mRNA isoforms with different transcription start sites are transcribed from the *Gm9999* locus on mouse chromosome (Chr) 7 and encode Kastor or Polluks. CDS, coding sequence; aa, amino acids. **c** Conservation of Kastor, Polluks, VDAC1, VDAC2, and VDAC3 among species analyzed by tBlastN. **d** RT-PCR analysis of the indicated mRNAs in adult mouse tissues. **e** In situ hybridization analysis of adult mouse testicular sections with riboprobes for Kastor or Polluks mRNAs. Nuclei were stained with nuclear fast red. Scale bars, 50 μm. **f, g** Immunoblot analysis of the testis (**f**) and spermatozoa collected from the cauda epididymis (**g**) of *Kastor*[FLAG/+] and *Polluks*[FLAG/+] adult mice was performed with antibodies to FLAG and to α-tubulin (loading control). **h** Immunoblot analysis of adult mouse tissues with antibodies to Kastor, to Polluks, or to HSP90 (loading control, short and long exposures for which are shown). Source data are provided as a Source data file.

(vimentin positive), with both polypeptides being detected only in germ cells at the step of spermiogenesis (Supplementary Fig. 2). Spermiogenesis is the final stage of spermatogenesis in which spermatids differentiate into mature spermatozoa. Spermatid development is classified into steps 1–16 on the basis of the morphology of the nucleus and acrosome[34]. Kastor was expressed during the middle steps of spermatid development (steps 6–11) and Polluks in the late steps (steps 14–16) (Fig. 2a). Kastor and Polluks were both found to be colocalized with TOM20, a specific marker of mitochondria (Fig. 2b). In mature spermatozoa, Polluks was also detected at the mitochondrial sheath, whereas Kastor was undetectable (Fig. 2c). These results indicated that the Kastor and Polluks polypeptides are both localized to mitochondria but are expressed at different steps of spermiogenesis.

**Kastor and Polluks localize to the OMM.** To validate the translation initiation sites of the identified ORFs, we generated expression constructs for full-length cDNAs of mouse Kastor or Polluks with a FLAG tag sequence inserted at the COOH-terminus of each ORF: Kastor-(C)FLAG and Polluks-(C)FLAG. We also deleted the putative initiating AUG codon of each ORF in-frame: Kastor-(C)FLAG/ΔATG and Polluks-(C)FLAG/ΔATG. The resulting constructs were introduced into HEK293T cells by transient transfection, and cell lysates were then examined by immunoblot analysis. Expression of Kastor or Polluks was not detected from the corresponding construct lacking the AUG codon (Fig. 3a), indicating that both Kastor and Polluks are indeed translated from the predicted initiation sites. Immunofluorescence analysis showed that Kastor-(C)FLAG and Polluks-(C)FLAG were also localized to mitochondria when expressed in HeLa cells (Fig. 3b).

Both Kastor and Polluks contain a hydrophobic domain in the NH$_2$-terminal region (Fig. 3c). Analysis with TMpred, a tool for prediction of transmembrane regions, suggested that both Kastor and Polluks possess a putative transmembrane region at the NH$_2$-terminus that is conserved in human and mouse (Fig. 3d). To examine whether Kastor and Polluks are simply attached to the mitochondrial membrane, as is the case for TIM44, or actually penetrate the membrane, as in the case of VDAC1, we performed topological analysis with mitochondrial fractions purified from HEK293T cells expressing Kastor-(C)FLAG or Polluks-(C)FLAG (Fig. 3e) as well as from the testis of *Kastor*[FLAG/+] and *Polluks*[FLAG/+] mice (in which the FLAG tag sequence is also inserted at the COOH-terminus of the respective polypeptide) (Fig. 3f). Ultrasonic treatment of the mitochondrial fraction followed by exposure to Na$_2$CO$_3$ in order to increase the pH resulted in disruption of mitochondrial structure, which was reflected by the disappearance of TIM44 from the membrane fraction and its appearance in the soluble fraction. However, such treatment did not affect the localization of VDAC1, Kastor-(C)FLAG, or Polluks-(C)FLAG in the membrane fraction of HEK293T cells or testis (Fig. 3e, f, and Supplementary Fig. 3a), suggesting that, like VDAC1, Kastor-(C)FLAG and Polluks-(C)FLAG substantially penetrate the mitochondrial membrane. Furthermore, treatment of the intact mitochondrial fraction of

HEK293T cells or testis with proteinase K in the absence of Triton X-100 resulted in the disappearance of the immunoblot signals corresponding to TOM20, Kastor-(C)FLAG, and Polluks-(C)FLAG (Fig. 3g, h, and Supplementary Fig. 3b), suggesting that the COOH-terminal regions of Kastor and Polluks are exposed to the cytosolic side of the OMM. Collectively, these results indicated that Kastor and Polluks are transmembrane proteins of the OMM and that their COOH-termini are directed toward the cytosol.

**Both Kastor and Polluks bind directly to VDAC.** To explore the potential function of Kastor and Polluks, we purified Kastor- or Polluks-containing protein complexes from the testis of *Kastor*[FLAG/+] or *Polluks*[FLAG/+] mice and from HEK293T cells transiently expressing Kastor-(C)FLAG or Polluks-(C)FLAG by immunoprecipitation with antibodies to FLAG (Fig. 4a). The precipitated proteins were subjected to liquid chromatography and tandem mass spectrometry (LC-MS/MS) with semi-quantitative analysis, which detected and further validated the existence of endogenous Kastor and Polluks polypeptides (Supplementary Fig. 4a, b, and Supplementary Data 2). Many proteins were commonly associated with both Kastor and Polluks (Supplementary Fig. 4c), and VDAC isoforms were prominent among these associated proteins (Fig. 4a, b, and Supplementary Data 3 and 4), consistent with the localization of Kastor, Polluks, and VDAC to the OMM. To validate the interaction of Kastor and Polluks with VDAC, we performed co-immunoprecipitation analysis with antibodies to FLAG. Although the association of all VDAC isoforms with Kastor-(C)FLAG or Polluks-(C)FLAG was confirmed in HEK293T cells (Fig. 4c), both Kastor-(C)FLAG and Polluks-(C)FLAG were preferentially associated with VDAC2 and VDAC3 in the testis of *Kastor*[FLAG/+] or *Polluks*[FLAG/+] mice (Fig. 4d). The specificity of the antibodies to each isoform of VDAC was validated in Neuro2A cells depleted of each isoform by transfection with a corresponding small interfering RNA (siRNA) (Supplementary Fig. 4d, e). The observation that binding of VDAC1 to Kastor-(C)FLAG or Polluks-(C)FLAG was virtually undetectable in the testis is likely explained by the fact that VDAC1 is exclusively localized in Sertoli cells in the testis[35].

To examine whether Kastor and Polluks interact directly with VDAC, we performed an in vitro binding assay with the use of purified recombinant mouse Kastor, Polluks, and VDAC3. *Acetabularia* rhodopsin I (ARI), a seven-transmembrane microbial rhodopsin, was examined as a negative control. Kastor-(C)FLAG or Polluks-(C)FLAG was incubated with VDAC3 or ARI in vitro and then subjected to immunoprecipitation with antibodies to FLAG. Whereas Kastor-(C)FLAG and Polluks-(C)FLAG interacted with ARI at only a low level, both polypeptides interacted prominently with VDAC3 (Fig. 4e), indicating that Kastor and Polluks bind directly to VDAC3.

**Loss of Kastor and Polluks severely impairs fertility.** We next generated mice deficient in either Kastor or Polluks, in both

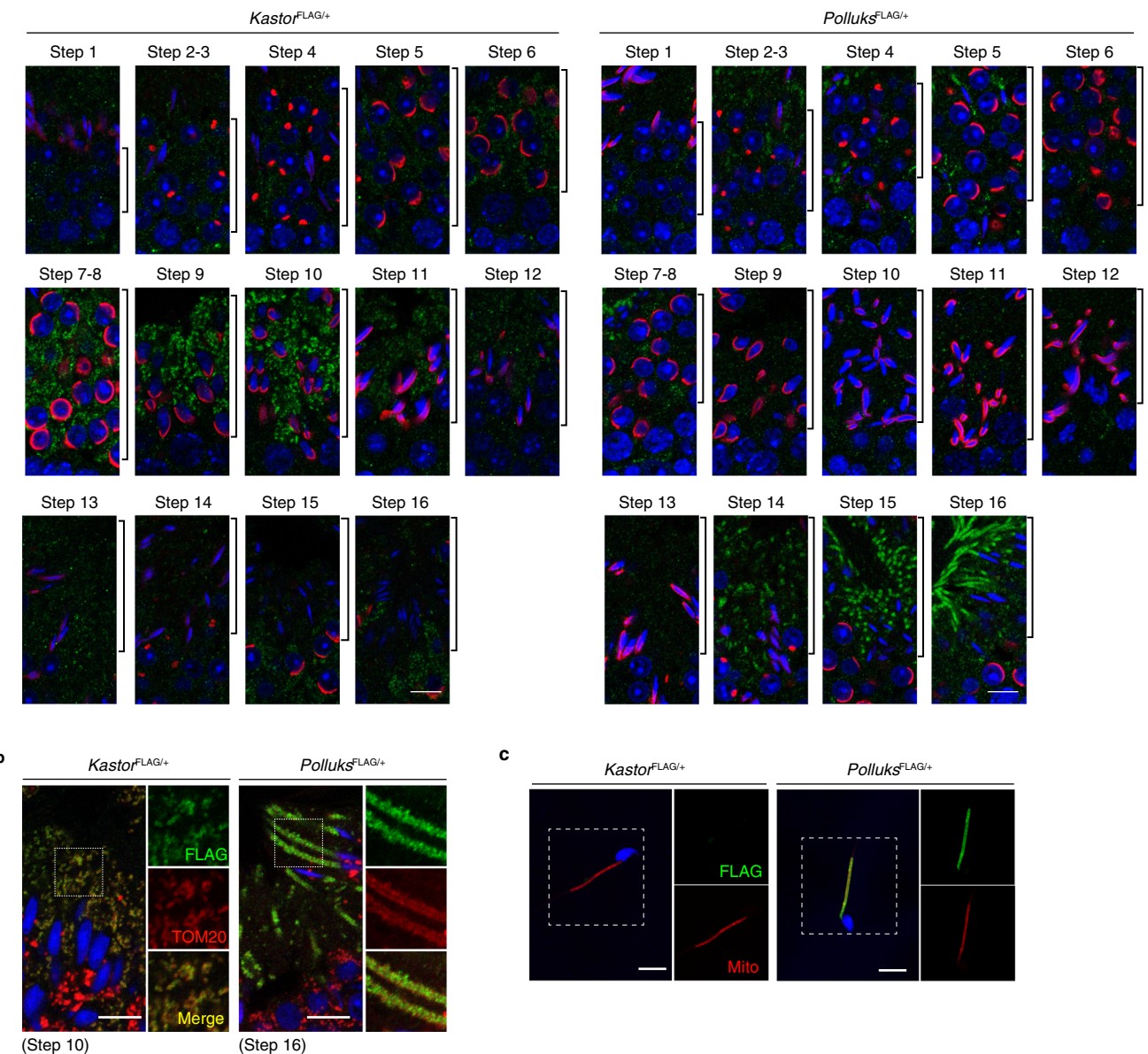

**Fig. 2 Kastor and Polluks are expressed at different steps of spermiogenesis. a** Immunofluorescence analysis of steps 1–16 of spermiogenesis in the testis of adult *Kastor*^FLAG/+ and *Polluks*^FLAG/+ mice. Each step is represented by the area indicated by the brackets. The steps were determined on the basis of the morphology of the nucleus and acrosome stained with 4′,6-diamidino-2-phenylindole (DAPI, blue) and Alexa Fluor–labeled peanut agglutinin (PNA, red), respectively. Kastor and Polluks were detected with antibodies to FLAG (green). Scale bars, 10 μm. **b**, **c** Immunofluorescence staining of the testis (**b**) and of spermatozoa collected from the cauda epididymis (**c**) of adult *Kastor*^FLAG/+ and *Polluks*^FLAG/+ mice with antibodies to FLAG. Mitochondria were detected with antibodies to TOM20 (**b**) or with MitoTracker Red (**c**), and nuclei were stained with DAPI (blue). The boxed areas in the left panel of each group are shown at higher magnification in the corresponding right panels. Scale bars, 20 μm (**b**) or 10 μm (**c**).

polypeptides, or in VDAC3 with the use of the CRISPR–Cas9 system (Supplementary Fig. 5a–d). Immunoblot analysis confirmed the absence of Kastor or Polluks, both poly-peptides, or VDAC3 in the testis or mature spermatozoa of the corresponding knockout (KO) mice: *Kastor*^−/− (Kastor KO), *Polluks*^−/− (Polluks KO), *Kastor*^−/−/*Polluks*^−/− (double KO [dKO]), and *Vdac3*^−/− (VDAC3 KO) mice (Fig. 5a). Of note, Kastor expression was upregulated by loss of Polluks, suggestive of a compensatory increase in the expression of the entire *Gm9999* locus in the Polluks KO mice (Fig. 5a).

Kastor KO, Polluks KO, and dKO mice were viable and manifested no substantial differences in body weight, testis weight or size, sperm count, or testicular histology compared with WT

mice (Fig. 5b–f). Most of these characteristics of VDAC3 KO mice were also similar to those of WT mice; whereas testicular weight was slightly reduced in the VDAC3 KO mice, a reduction in testis size was not obvious from gross appearance (Fig. 5b–f). VDAC3 KO male mice were infertile, consistent with a previous study[32], and breeding of male dKO mice with female WT mice revealed that only 2 of the 35 plugged females gave birth to pups, with the number of pups per mother also being low, indicative of severe infertility in the dKO male mice (Fig. 5g). Although both Kastor KO and Polluks KO male mice were fertile, the number of pups produced by WT females after breeding with the mutant males was significantly reduced compared with that for WT males (Fig. 5g). Given that loss of VDAC2 results in the almost

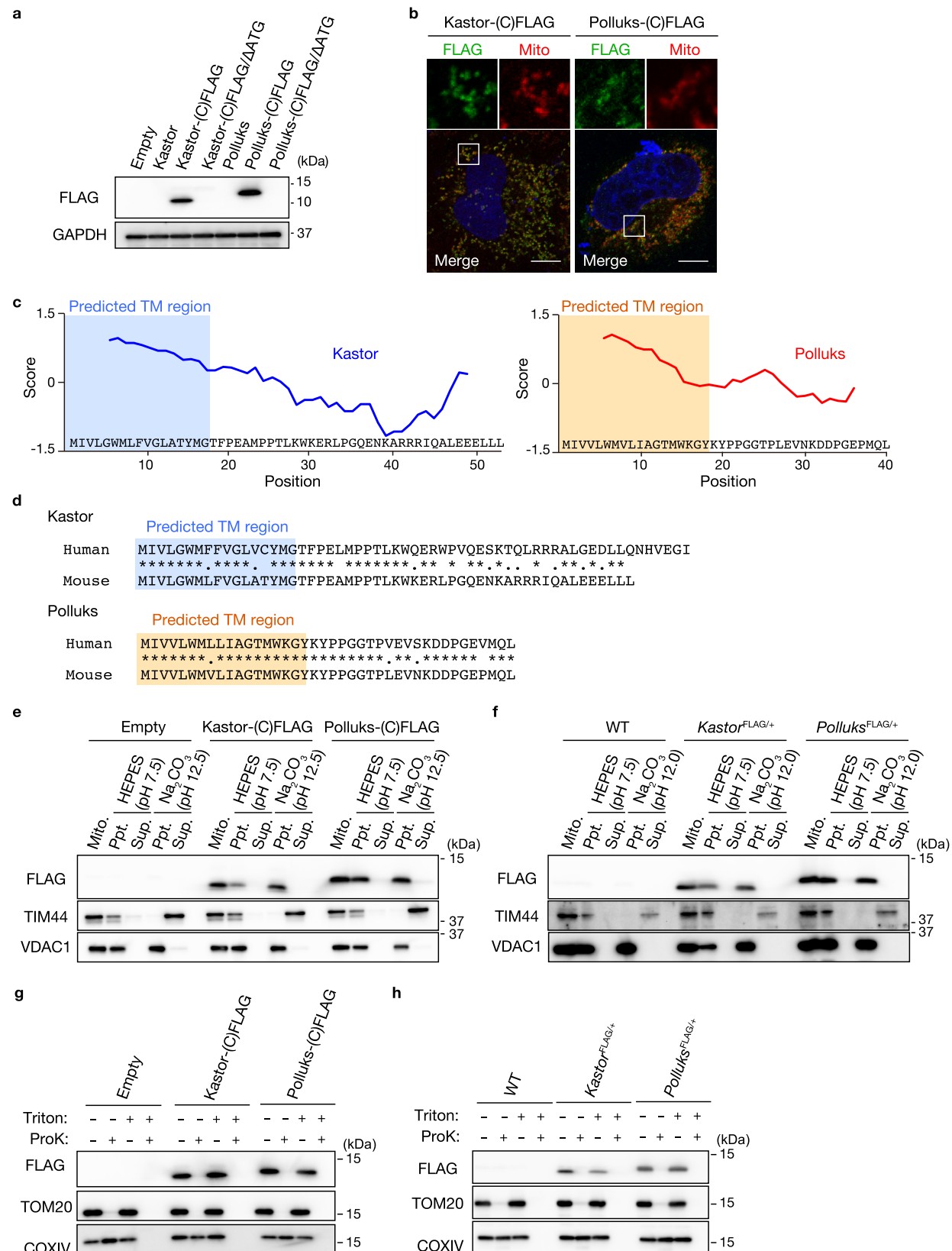

complete absence of spermatids and spermatozoa in male mice[31], the phenotype of dKO mice appeared more similar to that of VDAC3-deficient mice.

Computer-assisted spermatozoa analysis (CASA) revealed that the average path velocity (VAP), straight line velocity (VSL), and curvilinear velocity (VCL) were significantly decreased for sperm of Kastor KO, Polluks KO, dKO, and VDAC3 KO mice compared with those of WT mice (Fig. 5h–j). To examine whether spermatozoa from dKO or VDAC3 KO mice undergo capacitation and the acrosome reaction, we analyzed capacitation-associated tyrosine phosphorylation and the localization of IZUMO1 after incubation of the cells in capacitation

**Fig. 3 Kastor and Polluks are transmembrane polypeptides that localize to the OMM. a** Immunoblot analysis with antibodies to FLAG and to GAPDH (loading control) of HEK293T cells transiently transfected with the indicated vectors. **b** Immunofluorescence analysis of HeLa cells transiently expressing Kastor-(C)FLAG or Polluks-(C)FLAG with antibodies to FLAG. Mitochondria were detected with MitoTracker Red (Mito), and nuclei were stained with DAPI (blue). The boxed regions in the lower panels are shown at higher magnification in the corresponding upper panels. Scale bars, 10 μm. **c** Hydrophobicity scores of Kastor and Polluks determined by ProtScale. **d** Predicted transmembrane (TM) regions of human and mouse Kastor and Polluks analyzed by TMpred. **e, f** The mitochondrial fraction (Mito.) of HEK293T cells transiently expressing Kastor-(C)FLAG or Polluks-(C)FLAG (**e**) or of testis from $Kastor^{FLAG/+}$ and $Polluks^{FLAG/+}$ mice (**f**) was subjected to ultrasonic treatment, incubated with or without $Na_2CO_3$, and then centrifuged to yield a membrane fraction (ppt.) and a soluble fraction (sup.). These fractions were then subjected to immunoblot analysis with the indicated antibodies. **g, h** The mitochondrial fraction of HEK293T cells transiently expressing Kastor-(C)FLAG or Polluks-(C)FLAG (**g**) or of testis from $Kastor^{FLAG/+}$ and $Polluks^{FLAG/+}$ mice (**h**) was incubated in the absence or presence of proteinase K (ProK) and Triton X-100 and then subjected to immunoblot analysis with the indicated antibodies. Source data are provided as a Source data file.

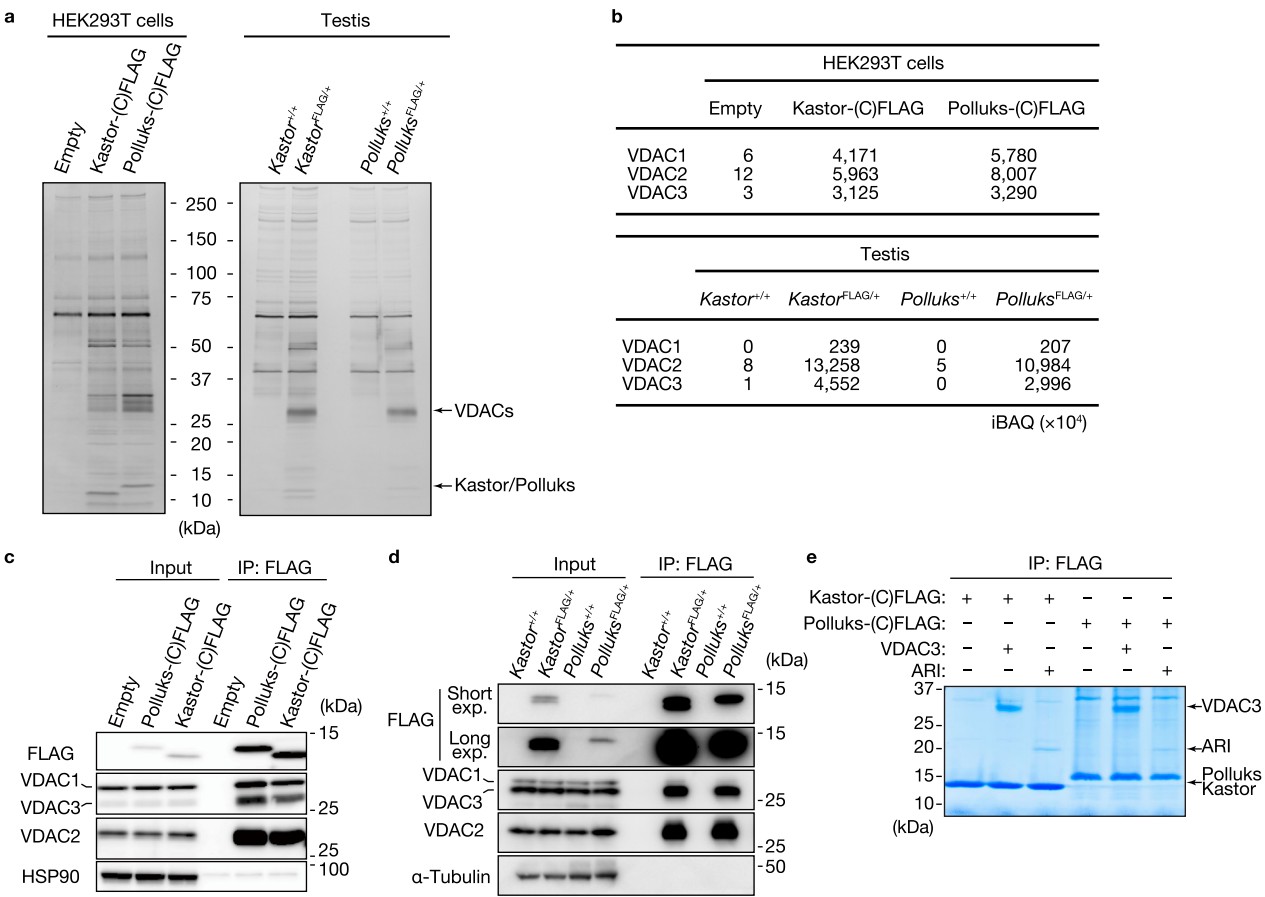

**Fig. 4 Both Kastor and Polluks bind to VDAC. a** Lysates either of HEK293T cells transiently expressing Kastor-(C)FLAG or Polluks-(C)FLAG or of the testis of adult $Kastor^{FLAG/+}$ or $Polluks^{FLAG/+}$ mice were subjected to immunoprecipitation with antibodies to FLAG, and the resulting precipitates were subjected to SDS–polyacrylamide gel electrophoresis (PAGE) and silver staining. **b** Intensity-based absolute quantification (iBAQ) of VDAC isoforms in immunoprecipitates prepared as in (**a**). **c, d** Immunoblot analysis with the indicated antibodies of immunoprecipitates (IP) prepared from HEK293T cells (**c**) or from mouse testis (**d**) as in (**a**). The original cell and tissue lysates (Input) were also analyzed. **e** In vitro binding assay for purified Kastor-(C)FLAG or Polluks-(C)FLAG with VDAC3 or ARI (negative control). FLAG immunoprecipitates isolated from the binding reaction mixtures were subjected to SDS-PAGE followed by staining with Coomassie brilliant blue. Source data are provided as a Source data file.

medium[36,37]. Both types of mutant spermatozoa manifested increased tyrosine phosphorylation and underwent the acrosome reaction to a similar extent as did WT cells (Supplementary Fig. 5e, f). We also incubated cumulus-free oocytes with spermatozoa in vitro, and found that spermatozoa of dKO or VDAC3 KO mice were capable of binding to the zona pellucida (ZP) (Supplementary Fig. 5g).

In vitro fertilization (IVF) analysis with cumulus-intact oocytes revealed a complete loss of fertilization capacity for spermatozoa from dKO or VDAC3 KO mice as well as a significant decrease in that for those from Kastor KO or Polluks KO mice (Fig. 5k).

Successful fertilization was apparent for all types of mutant spermatozoa when IVF was performed with oocytes from which the ZP had been removed (Fig. 5l). Together with the observation that several pups were obtained by breeding of male dKO mice with female WT mice (Fig. 5g), these results indicated that dKO spermatozoa are capable of fertilizing oocytes and producing pups.

Collectively, our findings suggested that spermatozoa from dKO or VDAC3 KO mice are able to fuse with oocytes by undergoing capacitation and the acrosome reaction but are unable to penetrate the ZP, likely as a result of a defect in sperm motility.

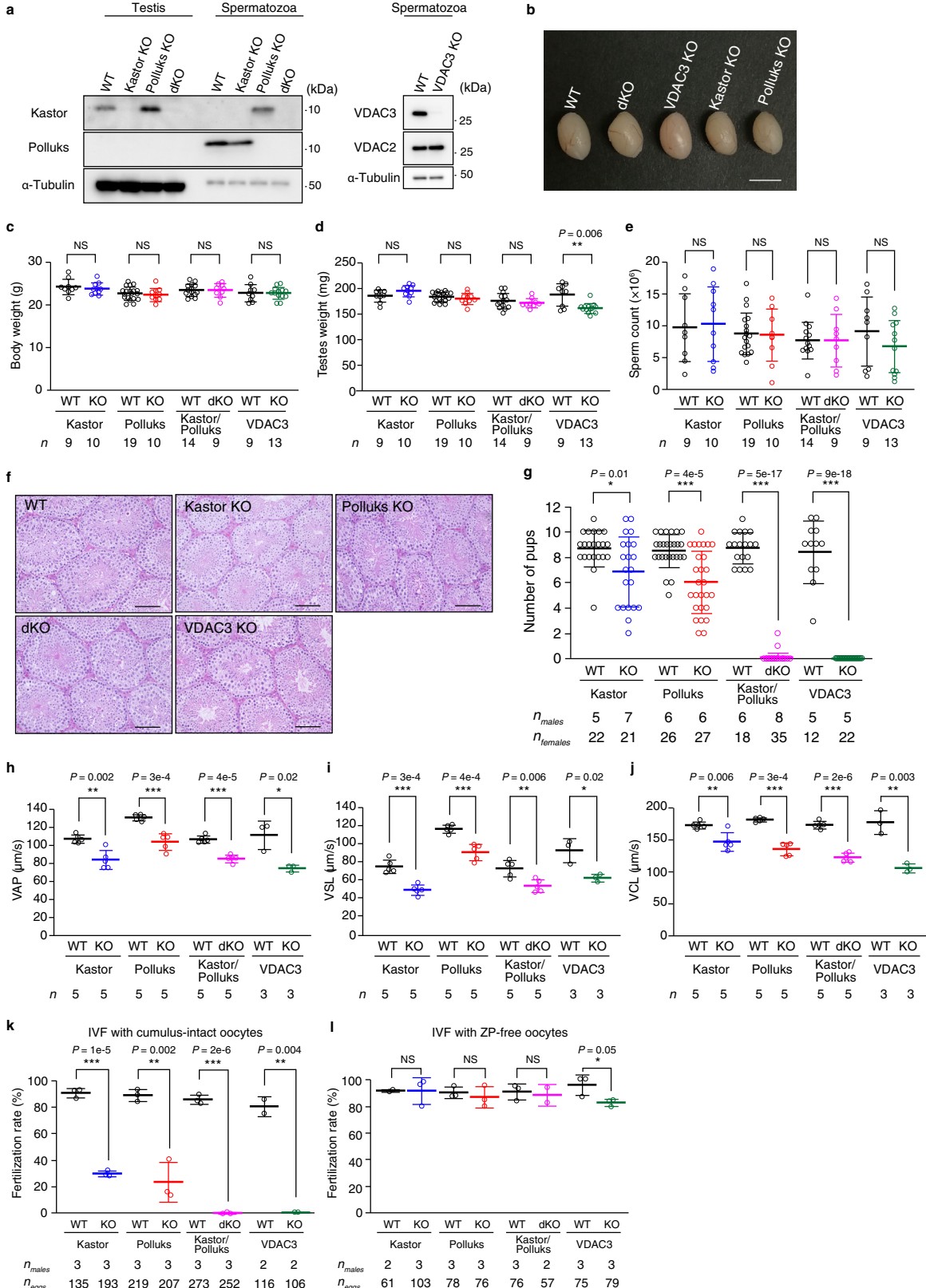

**Impaired mitochondrial sheath structure and wavy motility.** To elucidate the cause of the reduced sperm motility of dKO and VDAC3 KO mice, we first observed the morphology of mature spermatozoa from the cauda epididymis. We found that those from Kastor KO, dKO, or VDAC3 KO mice were sharply bent between

the midpiece and tail (Fig. 6a). The percentage of such abnormal bending was thus increased for spermatozoa from Kastor KO, dKO, and VDAC3 KO mice, but not for those from Polluks KO mice, compared with spermatozoa from WT controls (Fig. 6b). Such bending abnormalities have been found to disturb sperm motility and

**Fig. 5 Loss of Kastor and Polluks results in impaired male fertility. a** Immunoblot analysis of testis or of spermatozoa collected from the cauda epididymis of WT, Kastor KO, Polluks KO, dKO, or VDAC3 KO mice. **b** Gross appearance of the testes of WT and the indicated mutant mice at 8–10 weeks of age. Scale bar, 5 mm. **c–e** Body weight (**c**), testicular weight (**d**), and count for spermatozoa collected from the cauda epididymis (**e**) for the indicated numbers of WT and mutant mice at 8–10 weeks of age. **f** Hematoxylin-eosin staining of testicular sections of adult WT and the indicated mutant mice. Scale bars, 100 μm. **g** Number of pups produced by the indicated numbers of plugged WT female mice mated with the indicated numbers of male mice of the indicated genotypes. **h–j** Sperm motility analysis for adult mice of the indicated genotypes. Average path velocity (VAP) (**h**), straight line velocity (VSL) (**i**), and curvilinear velocity (VCL) (**j**) were analyzed with the CEROS or CEROS II systems. **k, l** Fertilization rate for an IVF assay performed with spermatozoa from the indicated numbers of adult mice of the indicated genotypes and the indicated numbers of either cumulus-intact (**k**) or ZP-free (**l**) oocytes from WT mice. All quantitative data are means ± s.d. *$P < 0.05$, **$P < 0.01$, ***$P < 0.001$; NS, not significant (unpaired two-tailed Student's $t$ test or Welch's $t$ test). Source data are provided as a Source data file.

to be present in spermatozoa with an abnormal morphology of the mitochondrial sheath, although the mechanism underlying such bending has remained unclear[14–16].

Given that transmission electron microscopy (TEM) has revealed morphological abnormalities in the mitochondrial sheath of VDAC3-null spermatozoa[32], we next examined the morphology and dynamics of sperm mitochondria during spermiogenesis in the testis by scanning electron microscopy (SEM), which allows visualization of mitochondrial sheath formation more clearly compared with TEM analysis[15,16]. Spermatids of dKO mice showed heterogeneity in mitochondrial size at the alignment step, a random direction of mitochondrial elongation at the interlocking step, and flattened mitochondria without coiling at the compaction step (Fig. 6c). Of note, mitochondria of VDAC3 KO spermatids manifested almost the same morphological defects as did those of dKO spermatids.

Kastor is expressed during the middle steps of spermatid development, whereas Polluks is expressed at the late steps (Fig. 2). Kastor KO spermatids manifested prominent abnormalities mainly at the alignment step—their mitochondria showed a nonuniform size at the alignment step, but they elongated in a uniform direction at the interlocking step, resulting in formation of a double-helical structure with nonuniformly sized mitochondria at the compaction step (Fig. 6c). In contrast, Polluks KO spermatids showed abnormalities at the interlocking and compaction steps, but not at the alignment step—their mitochondria were thus uniform in size at the alignment step but elongated in an irregular direction during the interlocking step and showed a flattened morphology without coiling at the compaction step (Fig. 6c). These abnormalities of Kastor KO and Polluks KO spermatids are thus consistent with the timing of the expression of the corresponding polypeptides.

We next monitored the movement of mature spermatozoa with a high-speed camera and quantified the bending angle of the midpiece (corresponding to mitochondria), excluding bent spermatozoa as seen in Fig. 6a. We found that the wavy motility of the midpiece apparent with WT spermatozoa was impaired in all mutant spermatozoa (Fig. 6d, e, and Supplementary Movies 1–5). VDAC3 KO spermatozoa showed the most pronounced extent of impaired flexion, followed by dKO spermatozoa, Polluks KO spermatozoa, and Kastor KO spermatozoa, a rank order consistent with the extent of mitochondrial sheath abnormality detected by SEM (Fig. 6c). These results suggested that the infertility of dKO and VDAC3 KO male mice is likely attributable to abnormal bending of spermatozoa and aberrant formation of the mitochondrial sheath, which result in impairment of wavy motility in the midpiece of these cells.

**VDAC3 KO spermatozoa manifest shortened mitochondrial sheath**. We also examined the morphology of mature spermatozoa from the cauda epididymis by SEM. Those of the various mutant mice showed an abnormal morphology similar to that observed at the compaction step in the testis (Fig. 7a).

However, only in VDAC3 KO spermatozoa, mitochondria near the neck and annulus were absent, resulting in a shorter mitochondrial sheath (Fig. 7a). We were not able to detect this shortening of the mitochondrial sheath in the testis by SEM for technical reasons. TEM further confirmed that mitochondria were not present near the neck and that outer dense fibers were exposed in this region of spermatozoa in the cauda epididymis of VDAC3 KO mice (Fig. 7b). To quantify these abnormalities, we performed MitoTracker staining analysis and found that most VDAC3 KO spermatozoa manifested a gap between the head and the mitochondrial sheath, which led to a shortening of the latter (Fig. 7c–e). These abnormalities were not observed in Kastor KO, Polluks KO, or dKO spermatozoa (Fig. 7c–e), suggestive of a Kastor/Polluks-independent function of VDAC3 in sheath formation.

**Mitochondrial metabolism and mPTP opening are unaffected**. VDAC regulates metabolic cross talk between mitochondria and the cytosol by controlling the influx or efflux of metabolites, cations, and nucleotides through the OMM[17], and VDAC1 deficiency in humans has been shown to impair the oxidation of pyruvate and the production of ATP[38]. We therefore performed metabolomics analysis of spermatozoa but found no differences in the levels of ATP or other metabolites related to the tricarboxylic acid cycle between dKO or VDAC3 KO mice and WT controls (Fig. 8a, b). Measurement of mitochondrial membrane potential with a fluorescent cationic carbocyanin dye (JC-1), as determined on the basis of the ratio of aggregates to monomers[39], also revealed that mitochondrial depolarization in dKO or VDAC3 KO spermatozoa did not differ from that in WT cells (Fig. 8c). To further evaluate mitochondrial metabolism, we measured the oxygen consumption rate (OCR) of spermatozoa, but again found that basal respiration, ATP production, maximal respiration, and spare capacity did not differ significantly between dKO or VDAC3 KO spermatozoa and WT spermatozoa (Fig. 8d–g). Together, these results thus suggested that the loss of Kastor/Polluks or VDAC3 had no marked impact on mitochondrial metabolism.

The mPTP allows passage of solutes with a size of up to 1.5 kDa at the inner mitochondrial membrane and is opened by $Ca^{2+}$ together with phosphate and reactive oxygen species (ROS)[17,40]. Opening of the mPTP results in collapse of the mitochondrial membrane potential, release of $Ca^{2+}$, cessation of ATP synthesis, and, eventually, mitochondrial swelling and cell death. Although the mPTP had been thought to be composed of VDAC, ANT, and cyclophilin D, genetic studies subsequently found that VDAC is not an essential component of the mPTP[41]. We therefore next evaluated mPTP opening with the use of the calcein-$CoCl_2$ fluorescence technique. $Co^{2+}$ quenches calcein fluorescence but is not able to penetrate the inner mitochondrial membrane when the mPTP is closed[42,43]. Indeed, a high mean fluorescence intensity (MFI) was observed in WT spermatozoa loaded with calcein acetoxymethyl ester (AM) and the MFI was decreased in spermatozoa exposed to

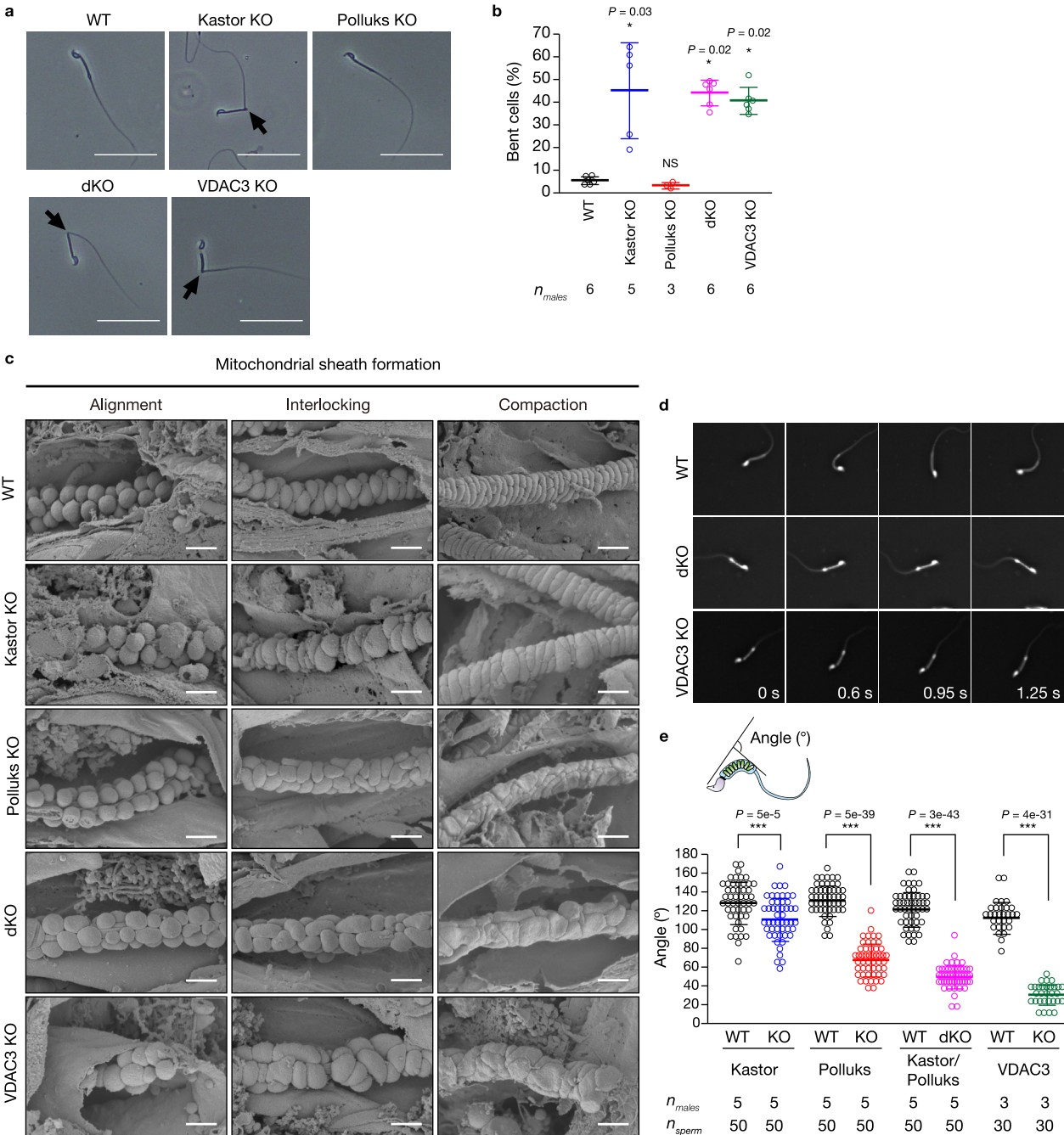

**Fig. 6 Abnormal mitochondrial sheath formation induced by loss of VDAC3 or of Kastor or Polluks. a, b** Gross appearance of spermatozoa (**a**) and quantification of the percentage of bent spermatozoa (**b**) collected from the cauda epididymis of adult WT, Kastor KO, Polluks KO, dKO, or VDAC3 KO mice. Arrows indicate the bent regions of spermatozoa. Scale bars, 50 μm. Quantitative data are means ± s.d. for the indicated numbers of male mice. **c** SEM images of mitochondrial sheath formation during spermiogenesis in the testis of mice of the indicated genotypes. Scale bars, 1.0 μm. **d, e** Images of wavy motion of spermatozoa obtained with a high-speed camera (**d**) and quantification of the bending angle of the midpiece (**e**). Quantitative data are means ± s.d. (10 spermatozoa per mouse, $n = 3$ or 5 mice per genotype). *$P < 0.05$, ***$P < 0.001$; NS, not significant (unpaired two-tailed Student's $t$ test, Welch's $t$ test or Steel's test). Source data are provided as a Source data file.

both calcein-AM and $CoCl_2$, whereas the fluorescence of calcein in the mitochondrial matrix was quenched by forced opening of the mPTP by treatment with the $Ca^{2+}$ ionophore A23187 (Fig. 8h). Monitoring of the MFI of calcein after incubation of dKO or VDAC3 KO spermatozoa with both calcein-AM and $CoCl_2$ revealed that mPTP opening status did not differ significantly between the mutant and WT spermatozoa (Fig. 8i, and Supplementary Fig. 6a). Furthermore, we monitored the $Ca^{2+}$ level in mitochondria with the fluorescent indicator Rhod2-AM, but we

found no significant difference between dKO or VDAC3 KO spermatozoa and WT control cells (Fig. 8j). In addition, neither ROS formation, as determined with the use of CellROX Orange[44,45], nor apoptosis was significantly altered in mature spermatozoa or the testis of dKO or VDAC3 KO mice compared with those of WT mice (Fig. 8k–m, and Supplementary Fig. 6b). Collectively, these results suggested that Kastor/Polluks and VDAC3 are not required for the regulation of mPTP opening or apoptotic signaling in spermatozoa.

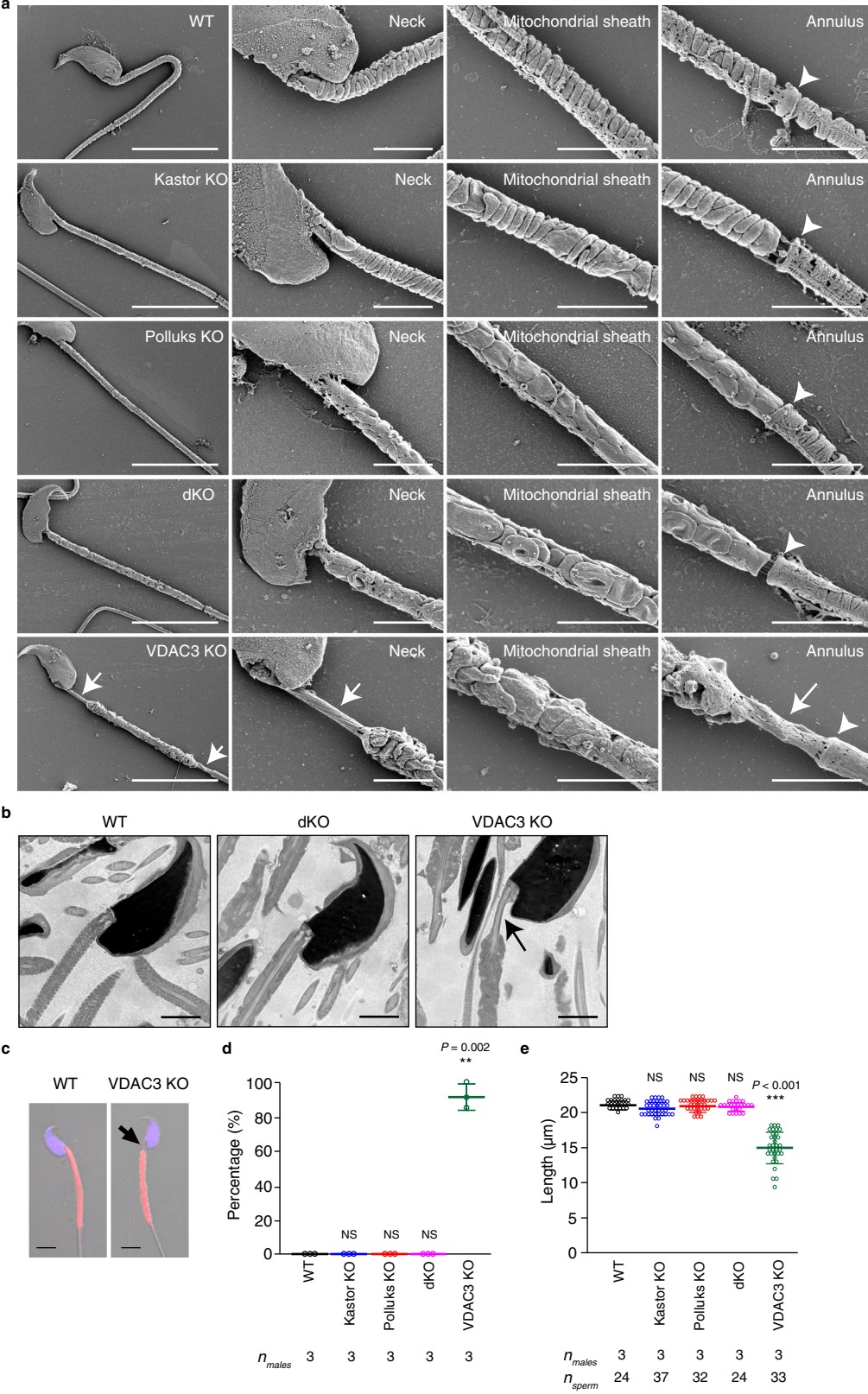

## Discussion

We have here identified Kastor and Polluks as polypeptides encoded by the mouse *Gm9999* locus, which was previously annotated as encoding a lncRNA. Although Kastor and Polluks have different amino acid sequences, both polypeptides were found to bind to VDAC in the OMM. The loss of Kastor gave rise to a nonuniform size of mitochondria and abnormal bending in spermatozoa, whereas the lack of Polluks resulted in irregular elongation and flattening of mitochondria without abnormal bending in these cells (Table 1). The bending and abnormal

**Fig. 7 Shortening of the mitochondrial sheath of VDAC3 KO spermatozoa. a** SEM of spermatozoa collected from the cauda epididymis of WT, Kastor KO, Polluks KO, dKO, and VDAC3 KO mice. Arrows indicate regions where mitochondria are absent. Arrowheads indicate the annulus. Scale bars, 10 μm (leftmost panels) or 4 μm (other panels). **b** TEM of spermatozoa in the cauda epididymis of WT, dKO, or VDAC3 KO mice. Arrow indicates a region where mitochondria are absent. Scale bars, 2 μm. **c–e** Fluorescence microscopic images of spermatozoa collected from the cauda epididymis (**c**) as well as quantification both of the percentage of spermatozoa with a gap between the head and the mitochondrial sheath (**d**) and of the length of the mitochondrial sheath (**e**). Mitochondria and nuclei were stained with MitoTracker Red and DAPI (blue), respectively. Arrow indicates a gap between the sperm head and mitochondrial sheath. Scale bars, 5 μm. Quantitative data are means ± s.d. **$P < 0.01$, ***$P < 0.001$ NS, not significant (unpaired two-tailed Student's $t$ test or Steel's test). Source data are provided as a Source data file.

structure of the mitochondrial sheath impaired the motility of spermatozoa, with the combination of these defects resulting in a greatly reduced fertility of dKO mice, similar to the case for VDAC3 KO mice (Table 1). Although we did not detect significant differences in mitochondrial metabolism or mPTP opening between dKO and WT spermatozoa (Table 1), the expression of Polluks in mature spermatozoa suggests that Polluks not only regulates sperm development but also may perform an unidentified function in the mature cells.

The structure of sperm mitochondria is highly variable among species. However, a mitochondrial helical structure is commonly observed in many mammalian species including human, rhesus monkey, mouse, rabbit, guinea pig, bull, and pig[46,47]. In contrast, in birds and fish, mitochondria are located at the midpiece of spermatozoa as in mammals, but they do not show a helical structure. For example, in spermatozoa of quail, mitochondria accumulate at the midpiece but manifest a morphology similar to that of those of mouse spermatozoa at the interlocking step[48], whereas, in cardinal fish, 7–10 spherical mitochondria are present in two rows[49,50]. Although VDAC3 is conserved from fish to mammals[25], Kastor and Polluks are conserved only in mammals, suggesting that the acquisition of Kastor and Polluks might have conferred the ability to form the mitochondrial helical structure in mammalian spermatozoa.

The morphology of the mitochondrial sheath is disorganized in spermatozoa lacking VDAC3 or both Kastor and Polluks, but it remains unclear how VDAC3 contributes to mitochondrial morphogenesis. Loss of armadillo repeat–containing 12 (ARMC12) results in an abnormal mitochondrial sheath structure, and ARMC12 interacts with VDAC2, VDAC3, TBC1 domain family member 21 (TBC1D21), and glycerol kinase 2 (GK2) in testicular germ cells[16]. Loss of TBC1D21 or GK2 also results in an abnormal mitochondrial sheath morphology[14–16], but the morphological abnormalities induced by loss of ARMC12, TBC1D21, or GK2 appear to differ from those observed in Kastor/Polluks-null spermatozoa, suggesting that these proteins might contribute to steps or mechanisms in VDAC3-dependent mitochondrial sheath formation different from those in which Kastor and Polluks play a role. Although loss of Kastor and Polluks largely recapitulated the phenotype of VDAC3 deficiency in this regard, only VDAC3-null spermatozoa showed shortening of the mitochondrial sheath (Table 1), suggestive of a Kastor/Polluks-independent function of VDAC3. Of note, immunoprecipitation and LC-MS/MS analysis of the testis of Kastor[FLAG/+] or Polluks[FLAG/+] mice identified not only VDAC2 and VDAC3 but also GK2, ARMC12, and TBC1D21 as binding proteins of both Kastor and Polluks (Supplementary Data 3). VDAC3 and its binding proteins including Kastor, Polluks, ARMC12, TBC1D21, and GK2 might thus constitute a complex in testicular germ cells that regulates various steps and mechanisms in mitochondrial sheath formation, although elucidation of the details of such regulation awaits further study.

We have identified Kastor and Polluks as polypeptides encoded by a testis-specific "lncRNA," but our PhyloCSF analysis suggested the existence of many other potential lncRNA-encoded polypeptides in testis (Supplementary Data 1). The comprehensive identification and

detailed analysis of such novel polypeptides encoded by hidden ORFs within lncRNAs will likely help to elucidate the molecular mechanisms of spermatogenesis and may provide a basis for the development of new therapies for infertility.

## Methods

**Animals.** All animal experiments were approved by the animal ethics committee of Kyushu University (A20-169-3) and were conducted in compliance with the university guidelines and regulations for animal experimentation. For generation of Kastor[−/−], Polluks[−/−], Kastor[−/−]/Polluks[−/−], Vdac3[−/−], Kastor[FLAG/+], and Polluks-[FLAG/+] mice, ribonucleoprotein (RNP) was prepared by mixing the CRISPR RNA (crRNA) and transactivating crRNA (tracrRNA) with recombinant Cas9 protein (Integrated DNA Technologies) (Supplementary Data 5). Single-stranded oligodeoxynucleotide (ssODN) homology repair templates were synthesized as 200-nt sequences (Supplementary Data 5). Mouse zygotes of the C57BL/6J strain were subjected to electroporation with each RNP with or without the corresponding ssODN. All mice were housed in the specific pathogen-free animal facility at Kyushu University in accordance with institutional guidelines under the following conditions: 22 °C ambient temperature, 50–60% humidity, 12 h dark/light cycle, and free access to water and rodent chow CA-1 (CLEA Japan). B6D2F1/Jcl (C57BL/6N Jcl × DBA/2N Jcl) female mice were used for the fertility test, and all other experiments were performed on the C57BL/6J background.

**Cell culture and transfection.** HEK293T (CRL-11268), HeLa (CCL-2), and Neuro2A (CCL-131) cells were obtained from American Type Culture Collection (ATCC) and were checked for mycoplasma contamination with the use of MycoAlert (Lonza). They were cultured under an atmosphere of 5% $CO_2$ at 37 °C in Dulbecco's modified Eagle's medium supplemented with 10% fetal bovine serum (Life Technologies) and antibiotics. For transient expression, cDNAs were subcloned into pcDNA3 (Invitrogen) with the use of a NEBuilder HiFi DNA Assembly Master Mix kit (New England BioLabs) and transfection was performed with the use of the X-tremeGENE 9 DNA Transfection Reagent (Sigma-Aldrich). For RNA interference, Neuro2A cells were transfected with 10 nM siRNAs (VDAC1, SASI_Mm02_00321250; VDAC2, SASI_Mm02_00321253; VDAC3, SASI_Mm01_00110006; or Universal Negative Control #1, SIC-001) with the use of the Lipofectamine RNAiMAX transfection reagent (Thermo Fisher Scientific).

**Antibodies.** Rat monoclonal antibodies to Kastor and to Polluks were generated with the use of glutathione S-transferase fusion proteins of mouse Kastor (residues 20–53) or Polluks (residues 1–40). Lymph nodes were collected from rats at 17 days after injection with the recombinant proteins, and lymphocytes isolated from the lymph nodes were fused with Sp2/0-Ag14 cells (CRL-1581, ATCC) with the use of polyethylene glycol 1500 (Roche) and then cultured in HAT (hypoxanthine-aminopterin-thymidine) medium. The resulting hybridomas were separated into single cells by limiting dilution, and monoclonal antibodies were obtained from the culture supernatants of the hybridomas. Antibodies to α-tubulin (13-8000) and to COXIV (A21348) were obtained from Thermo Fisher Scientific; those to FLAG (F1804 or F7425) and to phosphotyrosine (05-321) were from Sigma; those to DDDDK (PM020) for FLAG detection were from Medical & Biological Laboratories; those to VDAC2 (11663-1-AP) and to COXIV (11242-1-AP) were from ProteinTech; those to VDAC1/3 (ab14734), to VDAC1 (ab154856), to TIM44 (ab194829), and to TOM20 (ab186735) were from Abcam; those to TOM20 (sc-17764), to vimentin (sc-373717), to 3β-HSD (sc-515120), to SYCP3 (sc-74569), and to E-cadherin (sc-8426) were from Santa Cruz Biotechnology; those to IZUMO1 (73-045) were from BioAcademia; those to GAPDH (ADI-CSA-335-E) were from Enzo; and those to HSP90 (610419) were from BD Transduction Laboratories. The dilution factor used for each antibody is listed in Supplementary Data 5.

**RT and PCR or quantitative PCR (qPCR) analysis.** Total RNA extracted with the use of Isogen (Nippon Gene) was subjected to RT with the use of a QuantiTect Reverse Transcription Kit (Qiagen). The resulting cDNA was subjected to PCR analysis with specific primers (Supplementary Data 5), or was subjected to real-time PCR analysis with the use of Luna Universal qPCR Master Mix (New England BioLabs) and a StepOnePlus Real-Time system (Applied Biosystems). The amount of target mRNAs was normalized by that of β-actin mRNA.

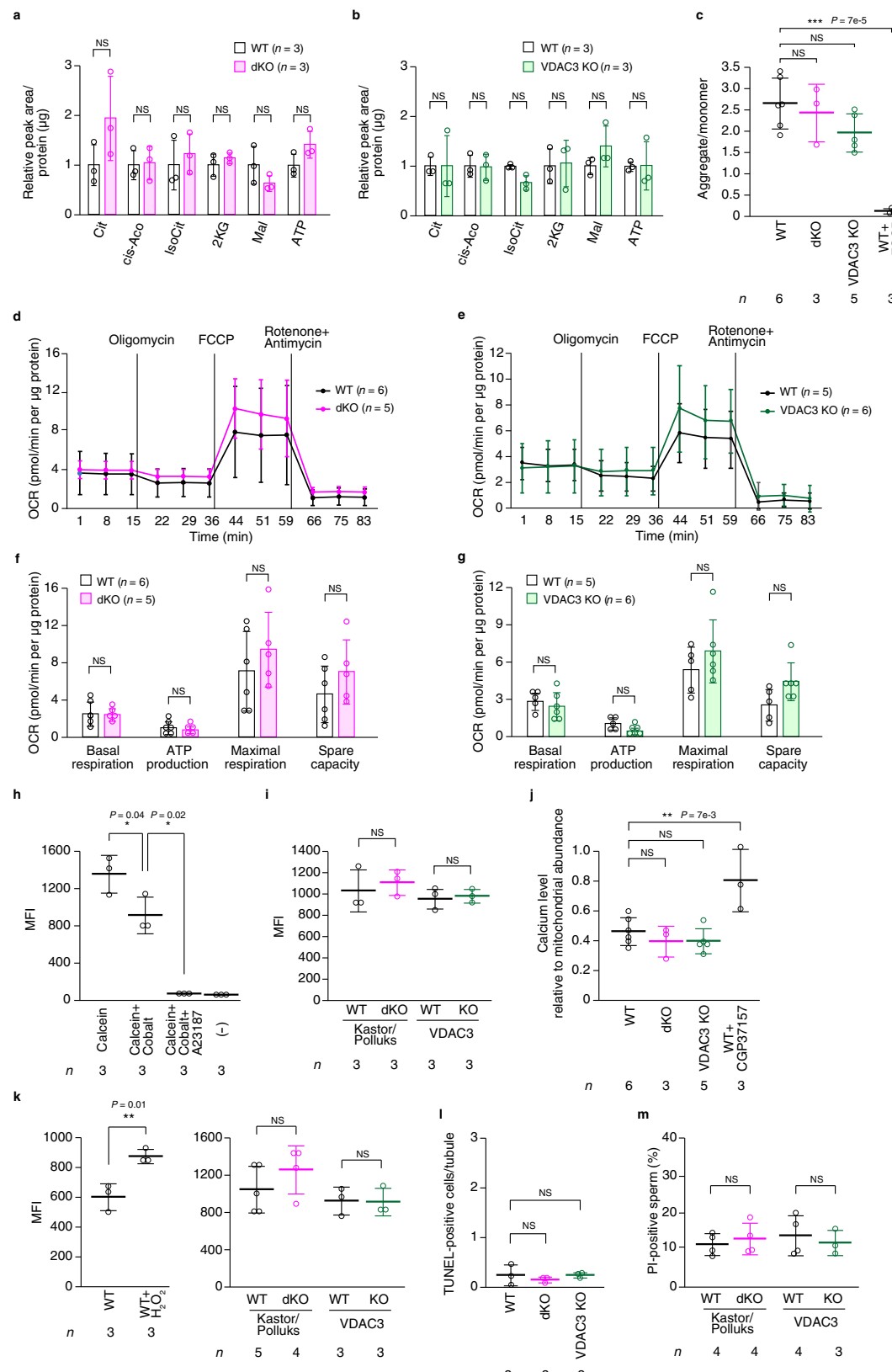

**Immunoblot analysis**. Protein samples were subjected to SDS-PAGE on 15% gels or 5–20% ExtraPAGE One Precast Gels (Nacalai Tesque). Membranes were incubated consecutively with primary antibodies and horseradish peroxidase-conjugated secondary antibodies (Promega), and signals were visualized with SuperSignal West Pico PLUS (Thermo Fisher Scientific) reagents and a ChemiDoc imaging system (Bio-Rad). Uncropped scans of the blots are shown in Source data and Supplementary Information.

**In vitro binding assay**. Modified His tag (MKDHLIHNHHKHEHAHAEH)–SUMO protease site–6×His–mouse VDAC3, modified His tag–TEV–Kastor–TEV–FLAG, and modified His tag–TEV–Polluks–TEV–FLAG recombinant proteins were synthesized in a cell-free synthesis reaction[51]. Purified Kastor or Polluks protein (20 µg) was bound to anti-FLAG M2 agarose beads (Sigma) by incubation for 90 min at 4 °C, after which 60 µg of VDAC3, or 60 µg of ARI as a negative control, were added to the beads and the binding mixture (final volume of 450 µl) was incubated for 120 min at 4 °C with

**Fig. 8 Lack of effect of VDAC3 or Kastor/Polluks ablation on mitochondrial metabolism and mPTP opening. a**, **b** Metabolome analysis of spermatozoa collected from the cauda epididymis of adult dKO (**a**) or VDAC3 KO (**b**) mice. Cit, citric acid; cis-Aco, cis-aconitic acid; IsoCit, isocitric acid; 2KG, 2-ketoglutaric acid; Mal, malic acid. **c** Mitochondrial membrane potential of spermatozoa collected from the cauda epididymis of adult mice of the indicated genotypes. Carbony cyanide 4-(trifluoromethoxy)phenylhydrazone (FCCP) was used as a control. **d**–**g** Oxygen consumption rate (OCR) traces for spermatozoa collected from the cauda epididymis of adult dKO (**d**) or VDAC3 KO (**e**) mice, as well as quantification of basal respiration, ATP production, maximal respiration, and spare capacity for the dKO (**f**) and VDAC3 KO (**g**) spermatozoa. **h**, **i** Analysis of mPTP opening in spermatozoa collected from the cauda epididymis of adult mice. Spermatozoa of WT mice were treated with the indicated agents (**h**), and those of dKO or VDAC3 KO mice were treated with calcein-AM and CoCl₂ (**i**). **j** Calcium ion concentration in mitochondria of spermatozoa collected from the cauda epididymis of adult mice of the indicated genotypes. **k** ROS formation in spermatozoa collected from the cauda epididymis of adult mice of the indicated genotypes. **l** Quantitative analysis of the number of TUNEL-positive cells per seminiferous tubule in testicular sections of adult mice of the indicated genotypes. **m** Quantitative analysis of the percentage of spermatozoa isolated from the cauda epididymas of adult mice that were positive for propidium iodide (PI) staining. All quantitative data are means ± s.d. for the indicated numbers of mice. *$P < 0.05$, **$P < 0.01$, ***$P < 0.001$; NS, not significant (paired two-tailed Student's $t$ test, unpaired two-tailed Student's $t$ test, one-way ANOVA, or two-way ANOVA). Source data are provided as a Source data file.

rotation. After four washes of the beads with a wash buffer (20 mM Tris-HCl [pH 8.0], 1 M NaCl, 0.1% *n*-dodecyl-*N*,*N*-dimethylamine-*N*-oxide [LDAO, Anatrace]), the bead-bound proteins were eluted with an elution buffer (20 mM Tris-HCl [pH 8.0], 1 M NaCl, 0.1% LDAO, 3×FLAG peptide [100 μg/ml, Sigma]) and subjected to SDS-PAGE followed by staining with Coomassie brilliant blue.

**Immunoprecipitation and proteome analysis.** Testis of *Kastor*^FLAG/+ or *Polluks*^FLAG/+ mice (at age of 10 to 56 weeks) or HEK293T cells transiently expressing FLAG-tagged Kastor or Polluks were lysed in lysis buffer (50 mM Tris-HCl [pH 7.5], 150 mM NaCl, 1% Triton X-100, cOmplete EDTA-free protease inhibitor cocktail [Roche]), the lysates were centrifuged at 20,400 × *g* for 20 min at 4 °C to remove debris, and the resulting supernatants were incubated for 90 min at 4 °C with anti-FLAG M2 affinity gel (Sigma). The resulting immunoprecipitates were washed three times with a wash buffer (50 mM Tris-HCl [pH 7.5], 150 mM NaCl, 0.1% Triton X-100), after which bead-bound proteins were eluted with wash buffer containing FLAG peptide (Sigma) at 0.5 mg/ml. The eluted proteins were subjected to SDS-PAGE followed by immunoblot analysis. For LC-MS/MS analysis, the eluates were fractionated by SDS-PAGE and subjected to silver staining. Protein bands were excised from the gel and subjected to in-gel digestion with trypsin, and the resulting peptides were dissolved in a solution comprising 0.1% trifluoroacetic acid and 2% acetonitrile for analysis with an LTQ Orbitrap Velos Pro mass spectrometer (Thermo Fisher Scientific). Acquired MS data were analyzed with MaxQuant and Mascot.

**Subcellular fractionation.** Testis of *Kastor*^FLAG/+ or *Polluks*^FLAG/+ mice (at age of 21 weeks) or HEK293T cells transiently expressing Kastor-(C)FLAG or Polluks-(C)FLAG were suspended in homogenization buffer (250 mM sucrose, 20 mM HEPES-NaOH [pH 7.5]) and homogenized with a Dounce homogenizer. The homogenate was centrifuged at 1000 × *g* for 10 min at 4 °C to remove debris, and the resulting supernatant was further centrifuged at 10,000 × *g* for 10 min at 4 °C. For sodium carbonate extraction, the resulting mitochondrial pellet was suspended in 20 mM HEPES-NaOH (pH 7.5) containing cOmplete EDTA-free protease inhibitor cocktail (Roche), subjected to ultrasonic treatment, incubated with 0.1 M Na₂CO₃ (pH 12.0 or 12.5) for 45 min on ice, and centrifuged at 100,000 × *g* for 30 min at 4 °C. The resulting supernatant and pellet were subjected to immunoblot analysis. For the proteinase K digestion assay, the original mitochondrial pellet was resuspended in assay buffer [150 mM NaCl, 10 mM Tris-HCl (pH 7.4)] and incubated for 20 min at 37 °C with proteinase K (final concentration, 0.8 mg/ml) in the absence or presence of 1% Triton X-100. The reaction was terminated by the consecutive addition of phenylmethylsulfonyl fluoride (final concentration, 2 mM) and SDS sample buffer before immunoblot analysis.

**In situ hybridization.** Riboprobes specific for mouse Kastor or Polluks mRNAs were synthesized from pcDNA3 vectors (Invitrogen) containing Kastor or Polluks cDNA with the use of a DIG RNA Labeling Kit (Roche). Testis (from mice at age of 10–14 weeks) was fixed with 4% paraformaldehyde in phosphate-buffered saline (PBS), embedded in paraffin, and sectioned at a thickness of 8 μm. The sections were depleted of paraffin, fixed again with paraformaldehyde, acetylated with triethanolamine (Sigma), permeabilized with PBS containing 0.3% Triton X-100, and incubated with the riboprobes overnight at 58 °C. They were then washed with saline sodium citrate (SSC), incubated overnight at 4 °C with alkaline phosphatase-conjugated Fab fragments to digoxigenin (Roche), washed with Tris-buffered saline containing 0.1% Tween 20 (TBST), and stained with 4-nitro blue tetrazolium (Roche) and 5-bromo-4-chloro-3-indolyl-phosphate (Roche). Nuclei were counterstained with nuclear fast red solution (ScyTek).

**Histology and immunohistofluorescence analysis.** Tissue (from mice at age of 10 to 56 weeks) was fixed with 4% paraformaldehyde in PBS. Sections with a thickness of 3 or 5 μm were subjected to hematoxylin-eosin staining or the TUNEL (terminal deoxynucleotidyl transferase-mediated dUTP nick-end labeling)

assay, respectively. The TUNEL assay was performed with the use of an Apoptosis in situ Detection Kit (Fujifilm Wako Pure Chemical). Nuclei were counterstained with methyl green solution. For immunofluorescence staining, OCT frozen sections (thickness of 5 μm) were treated with citrate buffer for antigen retrieval, washed with TBST, blocked with the use of a Mouse On Mouse Immunodetection Kit (Vector Laboratories), and incubated overnight at 4 °C in IHC buffer (0.3% Triton X-100, 0.01% azide, and 1% bovine serum albumin fraction V [BSA, Roche] in PBS) containing primary antibodies. The sections were washed three times with TBST and then incubated for 1 h at room temperature with Alexa Fluor-conjugated secondary antibodies (Thermo Fisher Scientific) in IHC buffer. For staining of the acrosome in testis, Alexa Fluor 647-conjugated PNA (Thermo Fisher Scientific) was added together with the secondary antibodies. The sections were mounted with the use of Vectashield Hard Set Mounting Medium with DAPI (Vector Laboratories) and observed with a Zeiss LSM700 microscope.

**Morphological analysis of spermatozoa.** Mature spermatozoa collected from the cauda epididymis (from mice at age of 8–17 weeks) were mounted on glass slides (Matsunami) and observed with an Olympus CKX53 light microscope for counting of bent spermatozoa.

**Immunofluorescence analysis of spermatozoa.** For monitoring of the acrosome reaction, spermatozoa collected from the cauda epididymis (from mice at age of 10 to 56 weeks) were cultured for 90 min in mHTF medium (Kyudo) with or without 20 μM A23187 (Sigma) for the last 30 min. For immunostaining of spermatozoa collected from the cauda epididymis of FLAG knock-in mice, the cells were cultured in mHTF medium for a few minutes to induce their dispersion and were then transferred to mHTF containing 100 nM MitoTracker Red CMXRos (Thermo Fisher Scientific) and cultured for 30 min. Spermatozoa were then spotted onto slides, air-dried, fixed with 4% paraformaldehyde for 10 min, exposed to ice-cold methanol for 5 min, and permeabilized with 0.5% Triton X-100 in PBS for 5 min. The slides were then washed three times with PBS, incubated first for 1 h at room temperature with 5% BSA and 0.1% Tween 20 in PBS and then overnight at 4 °C with primary antibodies, and washed another three times with PBS. They were finally incubated for 1 h at room temperature with Alexa Fluor-conjugated secondary antibodies, washed three times with PBS, and mounted with the use of Vectashield Hard Set Mounting Medium with DAPI (Vector Laboratories) before observation with a Zeiss LSM700 microscope.

**Immunofluorescence analysis of HeLa cells.** HeLa cells were cultured in medium containing 100 nM MitoTracker Red CMXRos (Thermo Fisher Scientific) for 30 min, fixed with 4% formalin, and incubated overnight at 4 °C in IF buffer (PBS containing 1% BSA and 0.3% Triton X-100) containing primary antibodies. They were washed three times with PBS, incubated for 1 h at room temperature in IF buffer containing Alexa Fluor 488-conjugated secondary antibodies, and mounted with the use of Vectashield Hard Set Mounting Medium with DAPI (Vector Laboratories) for observation with a Zeiss LSM700 microscope.

**Evaluation of mitochondrial membrane potential.** Spermatozoa collected from the cauda epididymis (from mice at age of 8 to 16 weeks) were cultured for 90 min in mHTF medium, transferred to 100 μl of mHTF containing 1 μl of JC-1 (Cayman) on a glass-bottom dish (Matsunami) coated with laminin (Corning), and cultured for 20 min. The mHTF containing JC-1 was replaced with mHTF containing DAPI (6 μg/ml), and the cells were observed with an Olympus IX70 microscope. The quantification of fluorescence values was performed with the use of ImageJ.

**Measurement of mitochondrial Ca²⁺ concentration.** Spermatozoa collected from the cauda epididymis (from mice at age of 8–16 weeks) were cultured for 2.5 h in mHTF medium, with or without 10 μM CGP37157 (Cayman) for the last

**Table 1 Summary of phenotypes of mutant mice.**

| | Kastor KO | Polluks KO | dKO | VDAC3 KO |
|---|---|---|---|---|
| Fertility and number of pups | Fertile but reduced number of pups | Fertile but reduced number of pups | Severely impaired fertility | Infertile |
| Proportion of bent spermatozoa | Increased | Normal | Increased | Increased |
| Spermatozoan mitochondrial morphology | Nonuniform size | Elongation in irregular directions and flattened shape | Nonuniform size, elongation in irregular directions, and flattened shape | Nonuniform size, elongation in irregular directions, and flattened shape |
| Gap between spermatozoan head and mitochondrial sheath | None | None | None | Yes |
| Length of mitochondrial sheath | Normal | Normal | Normal | Shortened |
| Spermatozoan metabolism | | | No significant differences | No significant differences |
| Spermatozoan mPTP opening and apoptosis | | | No significant differences | No significant differences |

60 min and with 0.1 μM Rhod2-AM (Abcam) and 0.1 μM MitoTracker Green FM (Thermo Fisher Scientific) for the last 30 min on a glass-bottom dish coated with laminin. The mHTF containing Rhod2-AM and MitoTracker Green FM was replaced with mHTF containing DAPI (6 μg/ml), and the cells were observed with an Olympus IX70 microscope. The quantification of fluorescence values was performed with the use of ImageJ.

**Measurement of dead spermatozoa.** Spermatozoa collected from the cauda epididymis (from mice at age of 8–12 weeks) were cultured in mHTF medium for a few minutes to induce their dispersion and were then transferred to mHTF on a glass-bottom dish coated with laminin and cultured for 30 min. The mHTF was replaced with mHTF containing propidium iodide (6 μg/ml), and the cells were observed with an Olympus IX70 microscope.

**Flow cytometric analysis of spermatozoa.** For ROS detection, mature spermatozoa collected from the cauda epididymis (from mice at age of 9–16 weeks) were incubated in mHTF medium for 90 min at 37 °C, with or without 0.006% $H_2O_2$ for the last 60 min and with 1 μM CellROX Orange (Invitrogen) for the last 30 min. Analysis of mPTP opening was performed with the use of a Mitoprobe Transition Pore Assay Kit (Molecular Probes). Mature spermatozoa collected from the cauda epididymis were incubated in mHTF medium for 2 h at 37 °C, with 0.03 μM calcein-AM, 4.8 μM $CoCl_2$, or 10 μM A23187 being added for the last 15 min. For removal of dead cells, spermatozoa were incubated with DAPI (6 μg/ml) for 5 min. Flow cytometric analysis was performed with a FACSVerse instrument (BD). The data were analyzed with the use of FlowJo V10 software.

**Measurement of the OCR of spermatozoa.** Mature spermatozoa were collected from the cauda epididymis (from mice at age of 8–15 weeks) in Tyrode's medium (Sigma) supplemented with 20 mM HEPES-NaOH (pH 7.5) (Gibco) and fatty acid–free BSA (Sigma) at 4 mg/ml. The spermatozoa ($2 \times 10^6$ per well) were transferred to a 24-well plate that had been coated with concanavalin A (Fujifilm Wako Pure Chemical) at 0.5 mg/ml (20 μl per well), and the plate was centrifuged at $1200 \times g$ for 2 min. The supernatant of each well was removed and replaced with 500 μl of Tyrode's medium, and the OCR was measured in real time with an XF24 extracellular flux analyzer (Seahorse Bioscience). The spermatozoa were exposed to 2 μM oligomycin for 19 min, to 1 μM FCCP for 19 min, and then to rotenone and antimycin (1 μM each) for 22 min. Spermatozoa were finally collected from each well for measurement of protein content and normalization of OCR values.

**Fertility test.** Two B6D2F1/Jcl (C57BL/6N Jcl × DBA/2N Jcl) female mice (at age of 8 weeks) (CLEA Japan) were caged with each test male mouse (at age of 8 to 10 weeks), and copulation was evaluated by checking each morning for a vaginal plug. The number of pups was determined as the number counted on the day of birth.

**IVF.** Oocytes collected from superovulating females (at age of 4 weeks) were placed in mHTF medium covered with paraffin oil. The eggs were then cultured with capacitated spermatozoa (from mice at age of 8–14 weeks) for 5 h, and the fertilization rate was determined at 24 h by counting the number of two-cell embryos. For removal of cumulus cells, oocytes were treated for 10 min with M2 medium (Sigma) containing hyaluronidase (Sigma) at 330 μg/ml. After removal of the cumulus cells, the ZP was removed by treatment with acidic Tyrode's solution (Sigma). ZP-free eggs were incubated for 6 h with spermatozoa at a density of $4 \times 10^4$/ml, after which the fertilization rate was determined by counting the number of eggs with pronuclei. For the ZP binding assay, cumulus-free eggs were incubated for 30 min with spermatozoa (from mice at age of 8–12 weeks) at a density of $2 \times 10^6$/ml, fixed with 4% paraformaldehyde, and observed with an Olympus IX70 microscope.

**Sperm motility analysis.** Sperm motility was measured with the use of a CEROS or CEROS II sperm analysis system. Spermatozoa of Kastor KO (at age of 10–12 weeks) and dKO (at age of 9–12 weeks) mice were examined with the CEROS system, and those of Polluks KO (at age of 11–14 weeks) and VDAC3 KO (at age of 13–18 weeks) mice with CEROS II. Sperm motility was also analyzed with an Olympus BX-53 microscope equipped with a high-speed camera (HAS-L1, Ditect) and was recorded at a frame rate of 200/s.

**SEM analysis of testis.** Testes (from mice at age of 9–18 weeks) dissected from mice were cut into 5-mm-thick slices with a safety razor, fixed for 1 h at 4 °C in 0.1 M phosphate buffer (pH 7.4) containing 1% $OsO_4$, immersed in 50% dimethyl sulfoxide, and then crushed with the use of a TF-2 apparatus (Eiko). The samples were washed with phosphate buffer, transferred to 0.1% $OsO_4$ in phosphate buffer, and incubated for 48 to 72 h at 20 °C. They were then fixed again for 1 h at 4 °C with 1% $OsO_4$ in phosphate buffer, stained for 2 h with 2% tannic acid (Fujifilm Wako Pure Chemical) and for 1 h with phosphate buffer containing 1% $OsO_4$, dehydrated with a graded series of ethanol solutions (50, 70, 90, 100, 100, and 100%), and subjected to critical point drying with the use of a Samdri-PVT-3D system (Tousimis). The specimens were finally mounted on sample stages, coated

with osmium with the use of an HPC-30W device (Vacuum Device), and observed with an S-4800 field-emission scanning electron microscope (Hitachi).

**SEM analysis of spermatozoa**. Spermatozoa collected from the cauda epididymis (from mice at age of 10–14 weeks) were incubated in Toyoda, Yokoyama, and Hoshi (TYH) medium to induce their dispersion, transferred to a 2.0-ml tube, and washed with 0.1 M phosphate buffer (pH 7.4). They were mounted on coverslips, fixed with 1% glutaraldehyde in phosphate buffer on ice, washed with phosphate buffer, fixed again with 1% $OsO_4$ in phosphate buffer containing 1% potassium ferrocyanide, and subjected to conductive staining with 1% tannic acid and 1% $OsO_4$. The specimens were dehydrated with a graded series of ethanol solutions, subjected to critical point drying with the use of a Samdri-PVT-3D system (Tousimis), coated with $OsO_4$ with an HPC-30W device (Vacuum Device), and observed with an S-4800 field-emission scanning electron microscope (Hitachi).

**TEM analysis of epididymis**. The epididymis dissected from mice (at age of 9–12 weeks) was immersed overnight in cacodylate buffer (pH 7.4) containing 2.5% glutaraldehyde, washed with PBS, fixed with 1% $OsO_4$ for 1 h on ice, washed with PBS, and dehydrated with a graded series of ethanol solutions (50, 70, 90, 95, 99.5, and 100%) before exposure first to ethanol:propylenoxide (1:1, v/v) and then to 100% propylenoxide. The tissue was then exposed consecutively for 30 min to propylenoxide:epoxy resin (1:1, v/v), for 30 min to propylenoxide:epoxy resin (1:2, v/v), overnight to pure epoxy resin, and for 1 h to DMP-30. It was then embedded in epoxy resin for 2 days at 65 °C, cut at a thickness of 80 nm with the use of a Leica EM UC7 device, and stained with 2% uranyl acetate for 5 min before examination with a JEM-1400 Plus electron microscope (JEOL) at 80 kV and image capture with a CCD Veleta 2 K × 2 K camera (Olympus).

**Metabolome analysis**. Caudal epididymal spermatozoa (from mice at age of 9–14 weeks) were incubated in mHTF medium for 2 h, collected by adsorption to a polytetrafluoroethylene membrane filter (pore size of 1.0 μm, Merck), and lysed by the addition of ice-cold methanol. The samples were vigorously agitated with a vortex-mixer for 1 min, subjected to ultrasonic treatment for 5 min, and maintained at −30 °C for 1 h. After addition of an internal standard (10-camphorsulfonic acid), the samples were mixed with chloroform, subjected to extraction by ultrasonic treatment for 5 min, and centrifuged at $16,000 \times g$ for 5 min at 4 °C. The upper phase was collected, mixed with chloroform and water, and centrifuged again at $16,000 \times g$ for 5 min at 4 °C. The upper phase was collected and subjected to ultrafiltration at $9100 \times g$ for 60 to 90 min at 4 °C with a UFC3LCCNB-HMT device (Human Metabolome Technologies), the filtrate was dried in an evaporator, and the resulting residue was dissolved in water. Metabolomics analysis of polar metabolites (organic acids and nucleotides) was performed by ion chromatography with a Dionex ICS-5000$^+$ HPIC system (Thermo Fisher Scientific) and Dionex IonPac AS11-HC-4 μm column (inner diameter of 2.1 mm, length of 150 mm, particle size of 4 μm; Thermo Fisher Scientific) followed by MS with a quadrupole-Orbitrap mass spectrometer (Thermo Fisher Scientific)[52].

**In silico analyses**. Orthologs of query proteins were identified by comparison of amino acid sequences for mouse with the use of tBlastN in National Center for Biotechnology Information (NCBI). Hydrophobicity scores of Kastor and Polluks were analyzed with ProtScale (https://web.expasy.org/protscale/), and predicted transmembrane regions of the polypeptides were analyzed with TMpred (https://embnet.vital-it.ch/software/TMPRED_form.html).

**Statistics and reproducibility**. Statistical significance of differences was calculated with the two-tailed paired or unpaired Student's $t$ test or Welch's $t$ test for unequal variance for comparisons between two groups, with one-way analysis of variance (ANOVA) followed by the Bonferroni post-hoc test or two-way ANOVA followed by the Sidak post-hoc test for comparisons among three or more groups, or with Steel's test for nonparametric multiple comparisons. Statistical calculations were performed with the use of JMP16 or Microsoft Excel. A $P$ value of <0.05 was considered statistically significant. The number of mice used in each experiment is indicated in the figure. The immunoblot, immunostaining and ZP binding experiments were performed at least three times with similar results. The images of electron microscopy were taken at several locations, and similar results were obtained.

**Reporting summary**. Further information on research design is available in the Nature Research Reporting Summary linked to this article.

## Data availability
The MS data have been deposited with the ProteomeXchange Consortium (http://proteomecentral.proteomexchange.org) via the JPOST partner repository under accession code PXD027126. Expression data of lncRNAs in testis were obtained from the NCBI database (https://www.ncbi.nlm.nih.gov). Source data are provided with this paper.

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

## Acknowledgements

We thank L. Cui, K. Ito, Y. Ishizuka-Katsura, and the Research Promotion Unit of the Medical Institute of Bioregulation at Kyushu University for technical assistance, as well as A. Ohta for help with preparation of the manuscript. We also thank Y. Ohkawa for the help in generating the monoclonal antibodies. This research was supported in part by grant JP21am0101082 from the Platform Project for Supporting Drug Discovery and Life Science Research (Basis for Supporting Innovative Drug Discovery and Life Science Research [BINDS]) of the Japan Agency for Medical Research and Development (AMED), as well as by KAKENHI grants from Japan Society for the Promotion of Science (JSPS) and the Ministry of Education, Culture, Sports, Science, and Technology of Japan to K.S. (JP20K16107), A.M. (JP20H05928 and JP20K21397), and K.I.N. (JP18H05215).

## Author contributions

S.M. performed experiments and wrote the manuscript. A.M. conceived and designed the project, performed experiments, and wrote the manuscript. K.S. performed experiments and wrote the manuscript. T.H., M.T., and C.S. performed experiments. K.I., H.S., and D.S. performed computational analysis. M. S., T.Y., T.I., Y.I., T.B., T.K.-S., M. Shirouzu, H.M., and M.I. supervised experimental design. K.I.N. coordinated the study and wrote the manuscript.

## Competing interests

The authors declare no competing interests.
