## [Peer Review File · Nature Communications]

Kastor and Polluks polypeptides encoded by a single gene locus cooperatively regulate VDAC and spermatogenesisReviewers' Comments:

Reviewer #1:

Remarks to the Author:

In this manuscript the authors address the putative role of two novel testis-specific polypeptides Kastor and Polluks (great names) in sperm function, linking their possible role with the known VDAC3 KO phenotype, thus suggesting a link with mitochondrial function, given this channels activity. The data suggests that, while the polypeptides are expressed at different stages of spermiogenesis, they both interact with VDAC in the outer mitochondrial membrane, and that this may lead to poor mitochondrial sheath formation, and poor hyperactivated motility, resulting in impaired male fertility.

The paper reports a very interesting set of extensive and labor-intensive observations using appropriate methodology, and the novelty of the two peptides coded by a sequence formerly considered to be a lncRNA is clear. However, there is no mechanistic approach as to how or why this is the case, and what the exact role of these polypeptides may be. Although this is not for lack of effort it remains a key issue when analyzing the data presented.

Some specific comments.

-Inserts of higher magnification of Fig 1e should be included, the image lacks definition (it can be done using a corner of the existing images). This is not offset by the data in Fig 2.

-The KO data in Fig 5 should be rewritten and organized including right ways the VDAC3 KO in order to flow better. Data mentioned on VDAC3 KO (histology, etc) should be included in Fig 5. It is not clear why some data is in Fig5, some is in S3. Notably, the smaller testis in the VDAC KO noted in S3 is not clear at all from the image in Fig5b.

Fig 5 should also include the velocity data, it is the only direct sperm functional analysis that seems relevant at this stage. Acrosome reaction or zona binding could, on the other hand be moved from Fig 5 (if needed) to S3, as they are not different.

-Still on motility, again the data on this shown in Fig 6 could be moved to Figure 5 (or the data reorganized) because there is an important issue to discuss in that the differences found in Fig6 do not seem enough to support those found in fertilization, pups, etc in Fig 5, notably comparing the single KOs with the dKO and VDAC KO. Looking at Fig 6c the Polluks KO seems equal to the dKO, with Kastor showing a mild effect, while in Fig 5 that is clearly not the case, and there Polluks and Kastor KO align, while the dKO phenotype is more severe. It is not linear how data in fig 6 translates to what was shown in Fig 5.

In fact that the Polluks KO shows a greater effect here (Fig 6), and that the dKO seems to not be very different, could be explained by the fact that, unlike Kastor, it is in fact present in mature sperm. But that contradicts the authors concept of mitochondrial sheath formation defects and suggests another functional role in mature sperm, that was not fully unveiled in previous steps.

-On the acrosome reaction: it is not immediately clear if this was spontaneous or induced. Both should be performed, the latter by either incubating with a trigger or ionophore.

-Sperm capacitation should be monitored (not just assumed by incubation in capacitation medium).

-ICSI should be performed to further validate the KO data in terms of motility defects.

-The data strongly suggests that these molecules may act via regulation of the mitochondrial transition pore, an important aspect of mitochondrial function that is not discussed. This could be evaluated (at least in transformed cells, if not in testis/sperm).

-A functional metabolic analysis on sperm (for example using the Seahorse or Oroboros systems) would also vastly improve the data at that level. The statement that metabolic effects are "unlikely" is not adequate, in my opinion.

-While apoptosis is an obvious issue to control for, so is ROS formation, and it is surprisingly absent.

-Mitochondrial defects seen by TEM (Fig 6) should be quantified somehow. Furthermore, again the data on mature sperm shown in S5 seems more relevant than the data in Fig6, as it shows clear potential consequences and a clear difference with the VDAC KO. This data also begs the question if sperm morphology in the different KOs (notably midpiece/tail defects) was monitored, and what the results were (should have been mentioned in Fig 5).

Minor

The text should be tweaked (especially in the Abstract). As the authors state in the Introduction (albeit in a convoluted manner) the focus is not lncRNAs of previous unknown function in the testis, but an erroneous annotation as a lncRNA of something that really isn't, and this is not the same thing.

Reviewer #2:

Remarks to the Author:

The manuscript submitted by Mise et al. showed two novel polypeptides (Kastor and Polluks) encoded by a single gene locus were essential for normal spermatogenesis.

In the manuscript the authors demonstrated these two polypeptides expressed in germ cells and inserted in the outer mitochondrial membrane and interact with VDAC to perform its function. In this study, the authors generated three conditional knockout mice and two knockin mice. The most important thing was to demonstrate/confirm Kastor and Polluks were encoded by Gm9999, a lncRNA. This reviewer read this manuscript carefully. The experimental design scheme is sound as well as logical and clear. To our knowledge, few studies were reported and demonstrated lncRNA encoded polypeptides, which impaired normal spermatogenesis and fertility and confirmed in the transgenic mice. So, this study was novel. I have some concerns that may improve the quality of the study.

1. Figure 1.h: Polluks was appeared in sperm but was not showed (expressed) in the testis in the figure. Sperm was also part of the testis. Why?
2. The authors showed that Kastor and Polluks are specifically expressed in testicular germ cells. However, germ cell included spermatogonia, spermatocytes, round sperm and sperm. The description of the expression of Kastor and Polluks should be detail and accurate. Also, I was wonder that if Kastor and Polluk were expressed in somatic cell, such as Leydig cells and Sertoli cells.
3. Whether the motility of sperm was affected in the result 5 (figure 5)? Because Kastor and Polluk were inserted with mitochondrial membrane that may cause the function of mitochondrial metabolism. Sperm count was provided in the study and the data of CASA should be showed. Because the figure 5 g showed no difference in the three groups.
4. Result 3 (figure 3) demonstrated that Kastor and Polluks are localized to OMM in 293T cells. However, the result in vivo also should be confirmed by IF.

Reviewer #3:

Remarks to the Author:

1. The authors have predicted and identified Kastor and Polluks as polypeptides encoded by a testis-specific lncRNA, which transcribes from the mouse Gm9999 locus high expressed in testis.
2. In this study, the two polypeptides Kastor and Polluks were found to bind to VDAC in the OMM and interact with VDAC, even they have different amino acid sequences. And the loss of both Kastor and Polluks results in formation of an abnormal mitochondrial sheath in spermatozoa and effect on the

wavy motion of the midpiece of spermatozoa that is similar to that apparent in VDAC3-null spermatozoa.

3.The comprehensive identification and deeply analysis of the two novel polypeptides encoded by hidden ORFs within lncRNAs in testis will likely help to elucidate the molecular mechanisms of spermatogenesis and may provide a basis for the development of new biomarkers or therapies for infertility.

4.The authors are encourage to provide the MS data of the identification of the two endogenous or high expressed polypeptides Kastor and Polluks, including the identified peptide list and sequence coverage.

5.This study is of great significance in the field of polypeptides encoded by lncRNA.

Reviewer #4:

None

Response to Reviewer #1

In this manuscript the authors address the putative role of two novel testis-specific polypeptides Kastor and Polluks (great names) in sperm function, linking their possible role with the known VDAC3 KO phenotype, thus suggesting a link with mitochondrial function, given this channels activity. The data suggests that, while the polypeptides are expressed at different stages of spermiogenesis, they both interact with VDAC in the outer mitochondrial membrane, and that this may lead to poor mitochondrial sheath formation, and poor hyperactivated motility, resulting in impaired male fertility.

The paper reports a very interesting set of extensive and labor-intensive observations using appropriate methodology, and the novelty of the two peptides coded by a sequence formerly considered to be a lncRNA is clear. However, there is no mechanistic approach as to how or why this is the case, and what the exact role of these polypeptides may be. Although this is not for lack of effort it remains a key issue when analyzing the data presented.

[Response] We thank the reviewer for the careful review of our manuscript and for the statements that “The paper reports a very interesting set of extensive and labor-intensive observations using appropriate methodology, and the novelty of the two peptides coded by a sequence formerly considered to be a lncRNA is clear.” We also thank the reviewer for many constructive suggestions that we feel have helped us to greatly improve our manuscript. Our specific responses to the points raised are as follows (this letter contains low-resolution thumbnails for clarity; please refer to the manuscript for high-resolution figures):

Specific comments:

-Inserts of higher magnification of Fig 1e should be included, the image lacks definition (it can be done using a corner of the existing images). This is not offset by the data in Fig 2.

[Response] In response to the reviewer’s comment, we have replaced the original images with higher magnification ones in the revised manuscript (**new Fig. 1e**). These improved in situ hybridization images clearly indicate that *Kastor* and *Polluks* are expressed at the middle and late steps of spermiogenesis, respectively, consistent with the results of Figure 2. We have now clarified this point in the revised manuscript (page 5, lines 132–133).

Mise et al. Figure 1

-The KO data in Fig 5 should be rewritten and organized including right ways the VDAC3 KO in order to flow better. Data mentioned on VDAC3 KO (histology, etc) should be included in Fig 5. It is not clear why some data is in Fig5, some is in S3. Notably, the smaller testis in the VDAC KO noted in S3 is not clear at all from the image in Fig5b. Fig 5 should also include the velocity data, it is the only direct sperm functional analysis that seems relevant at this stage. Acrosome reaction or zona binding could, on the other hand be moved from Fig 5 (if needed) to S3, as they are not different.

[Response] We apologize for the lack of logical flow in the structure of the original Figure 5. We have now substantially revised the data content and sequence in Figure 5 and their corresponding descriptions. Whereas we described VDAC3 KO phenotypes separately from dKO phenotypes in the original manuscript, these descriptions have now been combined with those for other mutant mice in the revised manuscript. In addition, most of the results including the velocity data shown in the original Supplementary Figure 3 have now been moved to the new Figure 5. The acrosome reaction and ZP binding data have been moved to the new Supplementary Figure 5, as suggested by the reviewer.

As pointed out by the reviewer, testicular weight for VDAC3 KO mice was slightly lower than that for WT mice. This small difference might actually be attributable to experimental error, given that the testes of WT controls for VDAC3 KO mice were slightly larger than those of WT controls corresponding to other KO mice. To address this point, we have now stated that “whereas testicular weight was slightly reduced in the VDAC3 KO mice, a reduction in testis size was not obvious from gross appearance” in the revised manuscript (page 8, lines 251–253).

-Still on motility, again the data on this shown in Fig 6 could be moved to Figure 5 (or the data reorganized) because there is an important issue to discuss in that the differences found in Fig6 do not seem enough to support those found in fertilization, pups, etc in Fig 5, notably comparing the single KOs with the dKO and VDAC KO. Looking at Fig 6c it the Polluks KO seems equal to the dKO, with Kastor showing a mild effect, while in Fig 5 that is clearly not the case, and there Polluks and Kastor KO align, while the dKO phenotype is more severe. It is not linear how data in fig 6 translates to what was shown in Fig 5. In fact that the Polluks KO shows a greater effect here (Fig 6), and that the dKO seems to not be very different, could be explained by the fact that, unlike Kastor, it is in fact present in mature sperm. But that contradicts the authors concept of mitochondrial sheath formation defects and suggests another functional role in mature sperm, that was not fully unveiled in previous steps.

[Response] Although we appreciate the reviewer’s suggestion about moving data from Figure 6 to Figure 5, it is technically not possible to do this because of space limitations (with Fig. 6c in particular being too large to be moved). However, we agree that Figures 5 and 6 are logically a single unit and should be described consecutively. We have thus amended the logical flow of the revised manuscript by changing the order of the data (**new Figs. 5 and 6**) as well as by adding new data for a VDAC3 KO–specific abnormality (**new Fig. 7**) and functional analysis of the mutant spermatozoa (**new Fig. 8**).

Given the importance of the reviewer’s point that the extent of the abnormality in mitochondrial morphology (**Fig. 6e**) does not seem to correlate with impairment of fertility (**Fig. 5g**), we further analyzed the phenotypes of these KO mice and found that the loss of Kastor, but not that of Polluks, gave rise to abnormal bending in a subset of spermatozoa (**new Fig. 6a, b**). This abnormal bending was also observed in the spermatozoa of dKO and

VDAC3 KO mice to the same extent as in those of Kastor KO mice (**new Fig. 6b**). Such bending abnormalities have been shown to disturb sperm motility and to be found in spermatozoa with abnormal mitochondrial morphology, although the mechanism responsible for this bending remains unclear (refs. 14–16). On the other hand, the bending angle of the midpiece during sperm motion was more adversely altered in Polluks KO mice than in Kastor KO mice (**Fig. 6 d, e**), which correlates with the extent of the abnormality observed in the SEM analysis of mitochondrial morphology during development (**Fig. 6c**).

Together, these results suggest that the causes of the reduced fertility in Kastor KO mice and Polluks KO mice may differ and are likely complex, as is now summarized in a **new Table 1** in the revised manuscript. The loss of Kastor resulted in a nonuniform mitochondrial size (**Fig. 6c**) and abnormal bending in spermatozoa (**new Fig. 6a, b**), whereas the lack of Polluks led to irregular elongation and flattening of mitochondria (**Fig. 6c**) and abnormal motion of spermatozoa (**Fig. 6 d, e**). We further explored a variety of possible functional abnormalities of Kastor/Polluks dKO spermatozoa, but we failed to find any significant differences in metabolomic characteristics (**Fig. 8a, b**), mitochondrial membrane potential (**Fig. 8c**), oxygen consumption rate (**new Fig. 8d–g**), mitochondrial permeability transition pore (mPTP) opening (**new Fig. 8h, i**), mitochondrial Ca^{2+} concentration (**Fig. 8j**), reactive oxygen species (ROS) production (**new Fig. 8k**), and apoptosis frequency (**Fig. 8l, m**) between the mutant and WT mice. As the reviewer points out, however, Polluks is expressed in mature spermatozoa, suggesting that Polluks may have additional functions in mature sperm that we did not uncover in the present study. We therefore now mention in the Discussion section of the revised manuscript (page 12, lines 401–405) that “Although we did not detect significant differences in mitochondrial metabolism or mPTP opening between dKO and WT spermatozoa (Table 1), the expression of Polluks in mature spermatozoa suggests that Polluks not only regulates sperm development but also may perform an unidentified function in the mature cells.” Collectively, our findings suggest that the combination of the various abnormalities observed in Kastor KO and Polluks KO mice results in severe impairment of fertility in dKO mice.

Mise *et al.* Figure 6

Mise *et al.* Figure 8

-On the acrosome reaction: it is not immediately clear if this was spontaneous or induced. Both should be performed, the latter by either incubating with a trigger or ionophore.

[Response] We apologize for the insufficient explanation for the acrosome reaction experiment. The data were for the spontaneous reaction. As suggested by the reviewer, we have now also analyzed the localization of IZUMO1 after incubation of spermatozoa in capacitation medium containing the Ca²⁺ ionophore A23187, and we found that both Kastor/Polluks-null and VDAC3-null spermatozoa also underwent the acrosome reaction to a similar extent as did WT cells in the induced condition (**new Supplementary Fig. 5f**). We have now addressed this point in the revised manuscript (page 9, lines 265–270).

Mise et al. Extended Data Figure 5

-Sperm capacitation should be monitored (not just assumed by incubation in capacitation medium).

[Response] To address the reviewer's request, we analyzed the tyrosine phosphorylation associated with capacitation after incubation of spermatozoa for 2 h in capacitation medium, given that protein phosphorylation, especially that on tyrosine residues, is a key event in this process (ref. 36). Both Kastor/Polluks dKO and VDAC3 KO spermatozoa manifested tyrosine phosphorylation to a similar extent as did WT cells (**new Supplementary Fig. 5e**), indicating that neither Kastor/Polluks nor VDAC3 is required for regulation of capacitation. We have now addressed this issue in the revised manuscript (page 9, lines 265–270).

Mise et al. Extended Data Figure 5

-ICSI should be performed to further validate the KO data in terms of motility defects.

[Response] We tried to perform intracytoplasmic sperm injection (ICSI) as suggested by the reviewer. ICSI with dKO spermatozoa was found to be technically difficult, however, given that separation of the head of these cells was almost impossible, probably as a result of their abnormal mitochondrial morphology, whereas the head of WT spermatozoa was readily separated by application of a few piezo pulses. Although it is possible to perform ICSI with whole spermatozoa, at least in our hands the fertilization rate was much lower and the associated error larger than for ICSI performed with the head only, rendering it difficult to compare dKO mice with WT controls.

However, we have shown that the dKO spermatozoa were able to fertilize oocytes without a ZP to a similar extent as did WT spermatozoa (**Fig. 5I**), and dKO male mice did produce a small number of pups (**Fig. 5g**). These results indicate that dKO sperm have the ability to fertilize an oocyte and that the zygotes are capable of producing pups. Even if ICSI were to be successfully performed with dKO sperm, the finding would only confirm the existing results that dKO sperm can produce pups. To clarify this point, we added the following sentence to the manuscript (page 9, lines 279–281): “Together with the observation that several pups were obtained by breeding of male dKO mice with female WT mice (Fig. 5g), these results indicated that dKO spermatozoa are capable of fertilizing oocytes and producing pups.”

Mise et al. Figure 5

-The data strongly suggests that these molecules may act via regulation of the mitochondrial transition pore, an important aspect of mitochondrial function that is not discussed. This could be evaluated (at least in transformed cells, if not in testis/sperm).

[Response] As suggested by the reviewer, we evaluated regulation of the mitochondrial permeability transition pore (mPTP) of spermatozoa with the calcein- CoCl_2 fluorescence technique. Co^{2+} quenches calcein fluorescence but cannot penetrate the mitochondrial inner membrane when the mPTP is closed, and CoCl_2 treatment therefore quenches calcein fluorescence in all intracellular compartments other than the mitochondrial matrix. Once the mPTP is open, the fluorescence of calcein in the mitochondrial matrix is also quenched, which allows determination of whether the mPTP is open or closed (ref. 42).

This technique has previously been applied to spermatozoa (ref. 43). A high mean fluorescence intensity (MFI) of calcein was observed in spermatozoa loaded with calcein acetoxymethyl ester (AM), but the MFI was lower in spermatozoa treated with both

calcein-AM and CoCl_2 , and the remaining fluorescence of calcein in the mitochondrial matrix was quenched by forced opening of the mPTP by exposure to the Ca^{2+} ionophore A23187 (**new Fig. 8h**). We monitored mPTP opening in Kastor/Polluks dKO and VDAC3 KO spermatozoa by incubation with both calcein-AM and CoCl_2 , but we detected no significant difference in MFI between the dKO or VDAC3 KO cells relative to WT controls (**new Fig. 8i**), suggesting that Kastor/Polluks and VDAC3 do not contribute to the regulation of mPTP opening. We have now addressed these points in the revised manuscript (pages 11–12, lines 365–381).

Mise et al. Figure 8

-A functional metabolic analysis on sperm (for example using the Seahorse ou Oroboros systems) would also vastly improve the data at that level. The statement that metabolic effects are "unlikely" is not adequate, in my opinion.

[Response] As suggested by the reviewer, we evaluated the metabolism of sperm mitochondria by measuring the oxygen consumption rate (OCR) of spermatozoa in real time with the use of an XF24 extracellular flux analyzer (Seahorse Bioscience). Mitochondrial respiration was monitored by measuring the OCR during sequential treatment with oligomycin, FCCP, and rotenone-antimycin. Monitoring of the OCR for dKO and VDAC3 KO spermatozoa revealed that basal respiration, ATP production, maximal respiration, and spare capacity did not differ from those of WT control cells (**new Fig. 8d–g**). Furthermore, metabolomics analysis revealed no significant differences between WT and either dKO or VDAC3 KO mice (**Fig. 8a, b**). These results provide further evidence that the loss of Kastor/Polluks or VDAC3 has no marked impact on mitochondrial metabolism. We have now addressed these points in the revised manuscript (page 11, lines 359–362).

Mise et al. Figure 8

-While apoptosis is an obvious issue to control for, so is ROS formation, and it is surprisingly absent.

[Response] To address the reviewer's concern, we stained spermatozoa with the ROS indicator CellROX Orange (ref. 44). We confirmed that treatment of spermatozoa with H_2O_2 increased the MFI of CellROX Orange and found that depletion of Kastor/Polluks or VDAC3 in spermatozoa did not affect ROS production (**new Fig. 8k**). We also determined apoptosis frequency for the mutant spermatozoa but failed to find any difference compared with WT cells (**Fig. 8l, m**). We have now addressed these points in the revised manuscript (page 12, lines 384–387).

Mise et al. Figure 8

-Mitochondrial defects seen by TEM (Fig 6) should be quantified somehow. Furthermore, again the data on mature sperm shown in S5 seems more relevant than the data in Fig6, as it

shows clear potential consequences and a clear difference with the VDAC KO. This data also begs the question if sperm morphology in the different KOs (notably midpiece/tail defects) was monitored, and what the results were (should have been mentioned in Fig 5).

[Response] Given that it is technically challenging to analyze a large number of spermatozoa by SEM analysis, we performed MitoTracker staining to quantify the abnormalities of mature spermatozoa detected by SEM. We found that most VDAC3 KO spermatozoa manifested a gap between the head and the mitochondrial sheath, which led to a shortening of the latter (**new Fig. 7c–e**). These abnormalities were not observed in Kastor KO, Polluks KO, or dKO spermatozoa, suggestive of a Kastor/Polluks-independent function of VDAC3. We have now addressed this issue in the revised manuscript (page 11, lines 341–346).

Mise et al. Figure 7

Minor:

The text should be tweaked (especially in the Abstract). As the authors state in the Introduction (albeit in a convoluted manner) the focus is not lncRNAs of previous unknown function in the testis, but an erroneous annotation as a lncRNA of something that really isn't, and this is not the same thing.

[Response] In response to this suggestion, we modified the text of the abstract and introduction in the revised manuscript to focus on peptide-coding RNAs that were misannotated as lncRNAs, rather than on lncRNAs of unknown function in the testis (pages 2–3, lines 33–57).

Response to Reviewer #2

The manuscript submitted by Mise et al. showed two novel polypeptides (Kastor and Polluks) encoded by a single gene locus were essential for normal spermatogenesis. In the manuscript the authors demonstrated these two polypeptides expressed in germ cells and inserted in the outer mitochondrial membrane and interact with VADC to perform its function. In this study, the authors generated three conditional knockout mice and two knockin mice. The most important thing was to demonstrate/confirm Kastor and Polluks were encoded by Gm9999, a lncRNA. This review read this manuscript carefully. The experimental design scheme is sound as well as logical and clear. To our knowledge, few studies were reported and demonstrated lncRNA encoded polypeptides, which impaired normal spermatogenesis and fertility and confirmed in the transgenic mice. So, this study was novel. I have some concerns may improve the quality of the study.

[Response] We thank the reviewer for the careful review of our manuscript and for the statements that “The experimental design scheme is sound as well as logical and clear” and that “To our knowledge, few studies were reported and demonstrated lncRNA encoded polypeptides, which impaired normal spermatogenesis and fertility and confirmed in the transgenic mice. So, this study was novel.” We also thank the reviewer for suggestions that we feel have helped us to greatly improve our manuscript. Our specific responses to the points raised are as follows (this response contains low-resolution thumbnails for clarity; please refer to the manuscript for high-resolution figures):

1. Figure 1.h: Polluks was appeared in sperm but was not showed (expressed) in the testis in the figure. Sperm was also part of the testis. Why?

[Response] This is likely attributable to a relatively low sensitivity of the monoclonal antibodies to Polluks as well as to a lower expression level of Polluks in the whole testis than in mature spermatozoa. Immunoblot analysis with anti-FLAG of lysates prepared from the testis of *Polluks*^{FLAG/+} mice confirmed that endogenous Polluks-(C)FLAG was indeed expressed in the testis (**Fig. 1f**). The monoclonal antibodies to Polluks were less sensitive than those to FLAG, and the sensitivity of the former antibodies was thus below the limit for detection of endogenous Polluks in the testis. Of note, a signal corresponding to Polluks was not detected with the monoclonal antibodies in mature spermatozoa of Polluks KO mice (**Fig. 5a**), confirming the high specificity of these antibodies. We have now clarified these points in the revised manuscript (page 5, lines 132–133).

2. The authors showed that Kastor and Polluks are specifically expressed in testicular germ cells. However, germ cell included spermatogonia, spermatocytes, round sperm and sperm. The description of the expression of Kastor and Polluks should be detail and accurate. Also, I was wonder that if Kastor and Polluk were expressed in somatic cell, such as Leydig cells and Sertoli cells.

[Response] We apologize for the inaccurate description of the expression of Kastor and Polluks. As suggested by the reviewer, we examined the localization of Kastor and Polluks in the testis of *Kastor*^{FLAG/+} and *Polluks*^{FLAG/+} mice by immunofluorescence analysis, and we found that neither Kastor nor Polluks was expressed in spermatogonia (E-cadherin positive) or spermatocytes (SYCP3 positive) or in Leydig cells (3 β -HSD positive) or Sertoli cells

(vimentin-positive) (**new Supplementary Fig. 2**). Together with the data in Figure 2, these new findings indicate that Kastor is expressed only during the middle steps of spermatid development (steps 6–11), whereas Polluks is expressed in the late steps of spermatid development (steps 14–16) and in mature spermatozoa. We have now addressed these points in the revised manuscript (pages 5–6, lines 151–159).

Mise *et al.* Extended Data Figure 2

3. Whether the motility of sperm was affected in the result 5 (figure 5)? Because Kastor and Polluk were inserted with mitochondrial membrane that may cause the function of

mitochondrial metabolism. Sperm count was provided in the study and the data of CASA should be showed. Because the figure 5 g showed no difference in the three groups.

[Response] In the original manuscript, the CASA data were included in Supplementary Figure 3, but, given that these data are key to our conclusion that abnormal mitochondrial morphology in VDAC3- or Kastor/Polluks-null spermatozoa leads to impaired motility, they have now been moved to a main figure in the revised manuscript (**Fig. 5h–j**).

Although we performed metabolomics analysis and detected no significant changes in the levels of metabolites related to the tricarboxylic acid cycle or ATP in dKO or VDAC3 KO spermatozoa compared with WT control cells in the original manuscript (**Fig. 8a, b**), we have now further evaluated the metabolism of sperm mitochondria by measuring the oxygen consumption rate (OCR) of spermatozoa in real time with the use of an XF24 extracellular flux analyzer (Seahorse Bioscience). Mitochondrial respiration was monitored by measuring the OCR during sequential treatment with oligomycin, FCCP, and rotenone-antimycin. Monitoring of the OCR of dKO and VDAC3 KO spermatozoa revealed that basal respiration, ATP production, maximal respiration, and spare capacity did not differ significantly from those of WT cells (**new Fig. 8d–g**). These results suggest that the loss of Kastor/Polluks or VDAC3 has no marked impact on mitochondrial metabolism. We have now addressed these points in the revised manuscript (page 11, lines 348–364).

Mise et al. Figure 5

Mise et al. Figure 8

4. Result 3 (figure 3) demonstrated that Kastor and Polluks are localized to OMM in 293T cells. However, the result in vivo also should be confirmed by IF.

[Response] It is technically difficult to determine whether Kastor and Polluks are localized in the OMM or IMM at the resolution of immunofluorescence analysis, or even with immuno-electron microscopy analysis. It is also challenging to elucidate the exact topology of Kastor and Polluks, such as whether they are simply attached to the mitochondrial

membrane or penetrate it. We therefore performed biochemical analysis of the testis from *Kastor*^{FLAG/+} and *Polluks*^{FLAG/+} mice similar to that we performed with transfected HEK293T cells (Fig. 3e, g) in order to determine the topology of the polypeptides in vivo. Mitochondrial fractions from the testis were exposed to Na₂CO₃ to increase pH, which did not affect the localization of Kastor-(C)FLAG and Polluks-(C)FLAG in the membrane fraction (**new Fig. 3f**). Furthermore, treatment of the mitochondrial fraction with proteinase K in the absence of Triton X-100 resulted in the disappearance of the immunoblot signals corresponding to Kastor-(C)FLAG and Polluks-(C)FLAG (**new Fig. 3h**). These results are completely consistent with those obtained with HEK293T cells expressing Kastor-(C)FLAG or Polluks-(C)FLAG, and they indicate that, also in vivo, Kastor and Polluks are transmembrane proteins of the OMM and that their COOH-termini are directed toward the cytosol. We have now addressed these points in the revised manuscript (pages 6–7, lines 185–205).

Mise et al. Figure 3

Response to Reviewer #3

1. The authors have predicted and identified Kastor and Polluks as polypeptides encoded by a testis-specific lncRNA, which transcribes from the mouse Gm9999 locus high expressed in testis.

2. In this study, the two polypeptides Kastor and Polluks were found to bind to VDAC in the OMM and interact with VDAC, even they have different amino acid sequences. And the loss of both Kastor and Polluks results in formation of an abnormal mitochondrial sheath in spermatozoa and effect on the wavy motion of the midpiece of spermatozoa that is similar to that apparent in VDAC3-null spermatozoa.

3. The comprehensive identification and deeply analysis of the two novel polypeptides encoded by hidden ORFs within lncRNAs in testis will likely help to elucidate the molecular mechanisms of spermatogenesis and may provide a basis for the development of new biomarkers or therapies for infertility.

4. The authors are encourage to provide the MS data of the identification of the two endogenous or high expressed polypeptides Kastor and Polluks, including the identified peptide list and sequence coverage.

5. This study is of great significance in the field of polypeptides encoded by lncRNA.

[Response] We thank the reviewer for the positive comments on our manuscript. The reviewer states that “This study is of great significance in the field of polypeptides encoded by lncRNA.” We also thank the reviewer for a suggestion, which we feel has helped us to improve our manuscript.

For comment 4, as suggested by the reviewer, we have now included the MS data for endogenous Kastor and Polluks, including the list of identified peptide sequences with their spectra and sequence coverage, as determined for the Kastor- or Polluks-containing protein complexes purified from the testis of *Kastor*^{FLAG/+} or *Polluks*^{FLAG/+} mice by immunoprecipitation with antibodies to FLAG (**new Supplementary Fig. 4a, b, and Supplementary Table 2**). These results provide further evidence for the presence of endogenous Kastor and Polluks polypeptides. We have now addressed these points in the revised manuscript (page 7, lines 211–214).

Mise et al. Extended Data Figure 4

Reviewers' Comments:

Reviewer #1:

Remarks to the Author:

The authors have adequately revised the manuscript, performing experiments and providing novel data on most of the points I raised initially, including the key aspects.

Although this new data adds some clarity to what could be the role of these polypeptides, as stated previously this is more of defining what that role is not, rather than what it may be. The mechanistic aspect remain elusive and that somewhat reduces the potential relevance of the work, as the authors themselves admit. However I have no further comments.

Reviewer #2:

Remarks to the Author:

All the suggestions had been revised. Thanks!

Reviewer #3:

Remarks to the Author:

The MS data for the identification of endogenous Kastor and Polluks, including the list of identified peptide sequences with their spectra and sequence coverage have been added in the revised manuscript. These results provide further evidence for the presence of endogenous Kastor and Polluks polypeptides. The revision is acceptable.